# 4-bit Shampoo for Memory-Efficient Network Training

**Sike Wang**
Beijing Normal University
sikewang@mail.bnu.edu.cn

**Pan Zhou**
Singapore Management University
panzhou@smu.edu.sg

**Jia Li**[†]
Beijing Normal University
jiali@bnu.edu.cn

**Hua Huang**
Beijing Normal University
huahuang@bnu.edu.cn

## Abstract

Second-order optimizers, maintaining a matrix termed a preconditioner, are superior to first-order optimizers in both theory and practice. The states forming the preconditioner and its inverse root restrict the maximum size of models trained by second-order optimizers. To address this, compressing 32-bit optimizer states to lower bitwidths has shown promise in reducing memory usage. However, current approaches only pertain to first-order optimizers. In this paper, we propose the first 4-bit second-order optimizers, exemplified by 4-bit Shampoo, maintaining performance similar to that of 32-bit ones. We show that quantizing the eigenvector matrix of the preconditioner in 4-bit Shampoo is remarkably better than quantizing the preconditioner itself both theoretically and experimentally. By rectifying the orthogonality of the quantized eigenvector matrix, we enhance the approximation of the preconditioner's eigenvector matrix, which also benefits the computation of its inverse 4-th root. Besides, we find that linear square quantization slightly outperforms dynamic tree quantization when quantizing second-order optimizer states. Evaluation on various networks for image classification and natural language modeling demonstrates that our 4-bit Shampoo achieves comparable performance to its 32-bit counterpart while being more memory-efficient[*].

## 1 Introduction

Deep neural networks (DNNs) have achieved great success in numerous fields, e.g., computer vision [20], natural language processing [38], and speech recognition [16]. A significant part of such success is attributed to first-order optimizers such as stochastic gradient descent with momentum (SGDM) [31] and AdamW [29]. Second-order optimizers, including K-FAC [30], Shampoo [18], AdaBK [41], CASPR [13], and Sophia [27], show great convergence properties, but often involve noticeable computation and memory costs. Anil et al. [2] provided several practical techniques for second-order optimizers to achieve substantial wall-clock time improvements over traditional first-order optimizers. The fast convergence property of second-order optimizers benefits from preconditioning the gradient with a matrix known as a preconditioner. The optimizer states for constructing the preconditioner and its inverse root can speed up optimization compared to first-order optimizers, but consume memory that could be used for model parameters, limiting the maximum model size trained within a given memory budget. With the increase in model size, the memory utilized by optimizer states can become a predominant factor in memory usage. This is the primary obstacle hindering the widespread use of second-order optimizers in the era of large models.

There are two main attempts to reduce memory consumed by optimizer states. Factorization uses low-rank approximation to optimizer states. This strategy has been applied to first-order optimizers [35, 3]

---

[†]Corresponding author.

38th Conference on Neural Information Processing Systems (NeurIPS 2024).

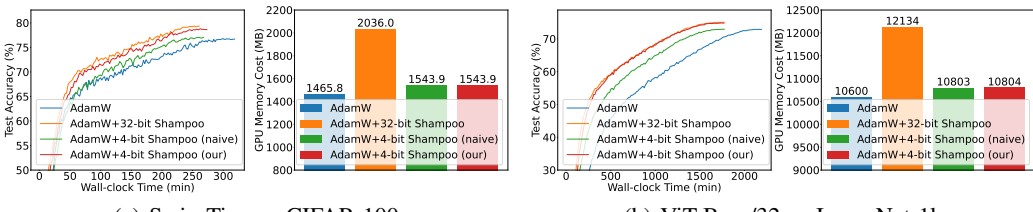

(a) Swin-Tiny on CIFAR-100        (b) ViT-Base/32 on ImageNet-1k

Figure 1: Visualization of test accuracies and total GPU memory costs of vision transformers. 4-bit Shampoo (naive) quantizes the preconditioner, while 4-bit Shampoo (our) quantizes its eigenvector matrix.

and second-order optimizers [14, 40]. In a comparable but distinct line of work, quantization utilizes low-bit to compress 32-bit optimizer states. Quantization is attractive due to its simplicity and wide applicability, which has been applied to first-order optimizers [8, 26]. Applying quantization to second-order optimizers poses a greater challenge, as first-order optimizers' states are elementwise, whereas second-order optimizers rely on matrix operations. To our knowledge, it has not been attempted before.

**Contributions:** In this paper, we present the first second-order optimizers with 4-bit optimizer states by taking Shampoo [18] as an example, while preserving the performance achieved with 32-bit optimizer states. While our focus is on Shampoo, we believe that our approach could also be applied to other second-order optimizers (see Table 4). Our main contributions are highlighted below.

Firstly, to maintain 32-bit performance, we propose quantizing the eigenvector matrix of a preconditioner in 4-bit Shampoo, rather than the preconditioner itself. The reason is that the small singular values of the preconditioner matter. Directly quantizing the preconditioner via block-wise quantization [8] at 4-bit precision can significantly alter the small singular values, leading to a drastic change in its inverse 4-th root and thus harming 4-bit Shampoo's performance. Quantizing the eigenvector matrix can help alleviate this issue, which is supported by experimental validation and theoretical insight. Additionally, with the eigenvector matrix, computing the inverse 4-th root is straightforward, ensuring that quantizing the eigenvector matrix does not lead to a rise in the total wall-clock time compared to quantizing the preconditioner (see Figure 1).

Secondly, we present two techniques for enhancing performance. As the eigenvector matrix of a preconditioner is orthogonal, we apply Björck orthonormalization [4] to rectify the orthogonality of the quantized eigenvector matrix, leading to improved approximation of preconditioner's eigenvector matrix and facilitating computation of its inverse 4-th root. Additionally, we observe that linear square quantization outperforms dynamic tree quantization [7] marginally when quantizing second-order optimizer states. The superiority of our developed 4-bit Shampoo is demonstrated in Figure 1.

Finally, we evaluate our 4-bit Shampoo on different image classification and natural language modeling tasks using convolutional neural network (CNN) and transformer architectures. Across all these benchmarks, our 4-bit Shampoo achieves similarly fast convergence comparable to its 32-bit counterpart, with no significant increase in losses for the trained models. Our 4-bit Shampoo uses less memory than its 32-bit counterpart, allowing for training of larger models with given resources.

## 2 Preliminaries

In this section, we present Shampoo and its implementation in our experiments. We also discuss quantization-based compression methods in a general formulation.

**Notations.** We use a non-bold letter like $a$ or $A$ to denote a scalar, a boldfaced lower-case letter like $\boldsymbol{a}$ to denote a vector, and a boldfaced upper-case letter such as $\boldsymbol{A}$ to denote a matrix. $\boldsymbol{u} = [u_i]^{\mathsf{T}}$ means that the $i$-th element of column vector $\boldsymbol{u}$ is $u_i$ and $\boldsymbol{U} = [\boldsymbol{u}_i]$ means the $i$-th column vector of matrix $\boldsymbol{U}$ is $\boldsymbol{u}_i$. Let $\boldsymbol{A}$ be a positive definite (PD) matrix and $s \in \mathbb{R}$, we define $\boldsymbol{A}^s = \boldsymbol{U}\boldsymbol{\Lambda}^s\boldsymbol{U}^{\mathsf{T}}$, where $\boldsymbol{U}\boldsymbol{\Lambda}\boldsymbol{U}^{\mathsf{T}}$ is the Singular Value Decomposition (SVD) of $\boldsymbol{A}$. $\mathrm{tr}(\boldsymbol{A})$ represents the trace of a matrix $\boldsymbol{A}$. The inner product of two matrices $\boldsymbol{A}$ and $\boldsymbol{B}$ is denoted as $\langle \boldsymbol{A}, \boldsymbol{B} \rangle = \mathrm{tr}(\boldsymbol{A}^{\mathsf{T}}\boldsymbol{B})$. The Frobenius norm of a matrix $\boldsymbol{A}$ is $\|\boldsymbol{A}\|_F = \sqrt{\langle \boldsymbol{A}, \boldsymbol{A} \rangle}$. $\boldsymbol{A} \odot \boldsymbol{B}$ means the elementwise matrix product (Hadamard product).

$\mathrm{Diag}(\boldsymbol{a})$ is a diagonal matrix with diagonal vector $\boldsymbol{a}$, while $\mathrm{diag}(\boldsymbol{A})$ means the diagonal vector of matrix $\boldsymbol{A}$.

## 2.1 Shampoo for Matrices

The update rule of Shampoo in the matrix case combined with a first-order optimizer $\mathcal{F}$ is

$$\mathbf{Shampoo}(\boldsymbol{W}_{t-1}, \boldsymbol{L}_{t-1}, \boldsymbol{R}_{t-1}, \boldsymbol{s}_{t-1}, \boldsymbol{G}_t) = \begin{cases} \boldsymbol{L}_t = \boldsymbol{L}_{t-1} + \boldsymbol{G}_t \boldsymbol{G}_t^{\mathsf{T}} \\ \boldsymbol{R}_t = \boldsymbol{R}_{t-1} + \boldsymbol{G}_t^{\mathsf{T}} \boldsymbol{G}_t \\ \widehat{\boldsymbol{G}}_t = \boldsymbol{L}_t^{-1/4} \boldsymbol{G}_t \boldsymbol{R}_t^{-1/4} \\ \widetilde{\boldsymbol{G}}_t = \widehat{\boldsymbol{G}}_t (\|\boldsymbol{G}_t\|_F / \|\widehat{\boldsymbol{G}}_t\|_F) \\ \boldsymbol{W}_t, \boldsymbol{s}_t = \mathcal{F}(\boldsymbol{W}_{t-1}, \boldsymbol{s}_{t-1}, \widetilde{\boldsymbol{G}}_t) \end{cases} \tag{1}$$

where $\boldsymbol{W}_t$ is the model parameters in matrix form, $\boldsymbol{L}_t$ and $\boldsymbol{R}_t$ are called preconditioners, $\boldsymbol{s}_t$ is the optimizer state of $\mathcal{F}$, and $\boldsymbol{G}_t$ is the gradient at $\boldsymbol{W}_{t-1}$. Note that $\boldsymbol{L}_t, \boldsymbol{R}_t, \boldsymbol{L}_t^{-1/4}$, and $\boldsymbol{R}_t^{-1/4}$ are PD matrices. The penultimate step in (1) is the grafting trick [1], which enables Shampoo to roughly apply the well-tuned learning rate schedule of $\mathcal{F}$. The optimization variable $\boldsymbol{W}_t$ does not represent all model parameters. It denotes a tensor of the model [18] or one block of a tensor [2]. In practice, we adopt an efficient and effective implementation of Shampoo for training DNNs following [2, 41] as described in Algorithm 4. In order to achieve efficient training, $\boldsymbol{L}_t, \boldsymbol{R}_t, \boldsymbol{L}_t^{-1/4}$, and $\boldsymbol{R}_t^{-1/4}$ are computed once every few hundred iterations. In this case, besides $\boldsymbol{L}_t$ and $\boldsymbol{R}_t$, their inverse 4-th roots should also be stored in memory, as computing them is computationally expensive. So training large models with Shampoo can be memory-intensive, consuming a significant amount of memory.

## 2.2 Quantization-based Compression Methods

Quantizing updated optimizer states using a quantizer and then dequantizing them with a dequantizer prior to use is an effective method for conserving memory. We focus exclusively on vectors, as tensors can be reshaped into vectors.

**Quantization.** According to the idea in [8, 26], a $b$-bit quantizer $\mathcal{Q}$ for $p$-dimensional real vectors is a mapping given by

$$\mathcal{Q} = (\mathcal{I} \circ \mathcal{N}, \mathcal{M}) : \mathbb{R}^p \to \mathbb{T}_b^p \times \mathbb{R}^p,$$

where $\mathcal{N}$ is a normalization operator on $\mathbb{R}^p$, $\mathcal{I}$ is an elementwise function mapping any real number to an element of $\mathbb{T}_b = \{0, 1, \ldots, 2^b - 1\}$, and $\mathcal{M}$ is a maximum operator on $\mathbb{R}^p$. For any $\boldsymbol{x} \in \mathbb{R}^p$, $\mathcal{N}$ and $\mathcal{M}$ satisfy $\mathcal{N}(\boldsymbol{x}) \odot \mathcal{M}(\boldsymbol{x}) = \boldsymbol{x}$.

A normalization operator $\mathcal{N}$ for $p$-dimensional vectors is a transformation on $\mathbb{R}^p$. It scales each element of a vector $\boldsymbol{x} \in \mathbb{R}^p$ into $[-1, 1]$. A block-wise normalization operator for a $p$-dimensional vector $\boldsymbol{x} = [x_1, x_2, \ldots, x_p]^{\mathsf{T}}$ is defined as

$$\mathcal{N}(\boldsymbol{x})_i = \frac{x_i}{\max_{j \in \mathbb{X}_i} \{x_j\}},$$

where $\mathcal{N}(\boldsymbol{x})_i$ is the $i$-th element of $\mathcal{N}(\boldsymbol{x})$, and $\mathbb{X}_i$ is a set satisfying $i \in \mathbb{X}_i \subset \{1, \ldots, p\}$. Usually, $\mathbb{X}_i$ should also satisfy $\mathbb{X}_i = \mathbb{X}_j$ or $\mathbb{X}_i \cap \mathbb{X}_j = \emptyset$ for $i, j \in \{1, \ldots, p\}$. In this case, for any $\boldsymbol{x} \in \mathbb{R}^p$, the number of different elements in $\mathcal{M}(\boldsymbol{x})$ is equal to the number of elements in set $\{\mathbb{X}_i | i = 1, \ldots, p\}$. Meanwhile, the number of the elements in $\mathbb{X}_i$ for any $i$ should be as close as possible to a value called block size.

The mapping $\mathcal{I}$ for $x \in \mathbb{R}$ in a $b$-bit quantizer $\mathcal{Q}$ is defined as

$$\mathcal{I}(x) = \underset{j \in \mathbb{T}_b}{\arg\min} |x - \mathcal{R}(j)|,$$

where $\mathcal{R}$ named quantization mapping is an elementwise function that maps any element in $\mathbb{T}_b$ into $[-1, 1]$, and $|\cdot|$ is the absolute operator for a scalar. There are three typical quantization mappings: linear quantization, dynamic quantization, and quantile quantization. Their specifications and visualizations can be found in [8].

**Dequantization.** Given a $b$-bit quantizer $\mathcal{Q} = (\mathcal{I} \circ \mathcal{N}, \mathcal{M})$ for a $p$-dimensional real vector $\boldsymbol{x} \in \mathbb{R}^p$, the corresponding dequantizer $\mathcal{D}$ is a mapping defined as

$$\mathcal{D}(\mathcal{Q}(\boldsymbol{x})) = \mathcal{D}(\mathcal{I} \circ \mathcal{N}(\boldsymbol{x}), \mathcal{M}(\boldsymbol{x})) = \mathcal{R}(\mathcal{I} \circ \mathcal{N}(\boldsymbol{x})) \odot \mathcal{M}(\boldsymbol{x}) : \mathbb{T}_b^p \times \mathbb{R}^p \to \mathbb{R}^p.$$

# 3 Methodology

In this section, we describe the design of our quantization-based compression method to realize 4-bit Shampoo with fast and high precision quantization. Let $\mathcal{Q} = (\mathcal{I} \circ \mathcal{N}, \mathcal{M})$ be a quantizer and $\mathcal{D}$ be its corresponding dequantizer as described in Subsection 2.2.

## 3.1 Quantizing the Eigenvector Matrices

A naive approach to realize 4-bit Shampoo is applying the compression methods proposed in [8, 26] to $\boldsymbol{L}_t$, $\boldsymbol{R}_t$, $\boldsymbol{L}_t^{-1/4}$, and $\boldsymbol{R}_t^{-1/4}$ in Shampoo (see (1)). A slightly improved approach is to quantize the four PD matrices excluding their diagonal elements, which typically much larger than their non-diagonal counterparts due to the non-negativity of the elements in $\mathrm{diag}(\boldsymbol{G}_t \boldsymbol{G}_t^{\mathsf{T}})$ and $\mathrm{diag}(\boldsymbol{G}_t^{\mathsf{T}} \boldsymbol{G}_t)$.

However, the naive approach can cause large quantization errors at 4-bit precision. This is because the quantization errors (or called perturbations) of quantizing $\boldsymbol{L}_t$ and $\boldsymbol{R}_t$ will transfer to $\boldsymbol{L}_t^{-1/4}$ and $\boldsymbol{R}_t^{-1/4}$. To verify this, we first introduce two criteria to evaluate the quantization errors of matrices. We do not use the elementwise criterion in [8]. Let $\boldsymbol{A}$ denote a 32-bit matrix, $g$ represent a transformation (can formed by quantization), and $f$ stand for a mapping, e.g., $f(\boldsymbol{A}) = \boldsymbol{A}^{-1/4}$. Then we define the normwise relative error (NRE) and angle error (AE) in $f$ of $g$ at $\boldsymbol{A}$ as

$$\mathrm{NRE} = \frac{\|f(\boldsymbol{A}) - f(g(\boldsymbol{A}))\|_F}{\|f(\boldsymbol{A})\|_F}, \quad \mathrm{AE} = \arccos\left(\frac{\langle f(\boldsymbol{A}), f(g(\boldsymbol{A}))\rangle}{(\|f(\boldsymbol{A})\|_F \|f(g(\boldsymbol{A}))\|_F)}\right).$$

We choose two PD matrices of order 1200. The first one $\boldsymbol{A}_1$ is derived from the real world. It is a preconditioner in 32-bit Shampoo combined with AdamW for training a Swin-Tiny model. The second one $\boldsymbol{A}_2 = \boldsymbol{U}\boldsymbol{\Lambda}\boldsymbol{U}^{\mathsf{T}}$ is synthetic, constructed from a random orthogonal matrix $\boldsymbol{U}$ and a diagonal matrix $\boldsymbol{\Lambda}$ with only two distinct diagonal values. Table 1 shows the quantization errors in $f(\boldsymbol{A}) = \boldsymbol{A}^{-1/4}$ of the naive approach at these two matrices, which are remarkably high. More analyses are given in Appendix D. The key point is that the singular values of $\boldsymbol{A}_i (i = 1, 2)$ follow a specific distribution (see Figure 2). In this scenario, a slight perturbation of $\boldsymbol{A}_i$ will significantly alter its small singular values, resulting in a drastic change to $\boldsymbol{A}_i^{-1/4}$.

To address this issue, we propose quantizing the eigenvector matrix of a preconditioner in Shampoo, rather than the preconditioner itself. Namely, a preconditioner $\boldsymbol{A}$ is a PD matrix, and its SVD is $\boldsymbol{U}\boldsymbol{\Lambda}\boldsymbol{U}^{\mathsf{T}}$, where $\boldsymbol{U}$ represents the eigenvector matrix and $\boldsymbol{\Lambda}$ denotes the singular value matrix. Given that $\boldsymbol{\Lambda}$ is a diagonal matrix, we can focus on quantizing $\boldsymbol{U}$ using $\mathcal{Q}$ while leaving $\boldsymbol{\Lambda}$ unchanged. From Table 1, one can observe that quantizing $\boldsymbol{U}$ can significantly reduce the quantization errors. We will theoretically discuss the advantages of quantizing $\boldsymbol{U}$ compared to quantizing $\boldsymbol{A}$ in Section 4. In practice, the randomized SVD method [19] is adopted to compute the SVD of $\boldsymbol{A}$ efficiently, as shown in [40]. We want to highlight that quantizing the original $\boldsymbol{L}_t$ and $\boldsymbol{R}_t$ in Shampoo involves significant computational burdens to compute their inverse 4-th roots $\boldsymbol{L}_t^{-1/4}$ and $\boldsymbol{R}_t^{-1/4}$, whereas quantizing the eigenvector matrices of $\boldsymbol{L}_t$ and $\boldsymbol{R}_t$ allows for rapid inverse root calculation. So the computational time required for both approaches is comparable (see Figure 1).

Table 1: Quantization errors in $\boldsymbol{A}^{-1/4}$ of different quantization schemes at a PD matrix $\boldsymbol{A}$. We employ block-wise normalization with a block size of 64. $\boldsymbol{U}$ is the eigenvector matrix of $\boldsymbol{A}$, QM = quantized matrix, and OR = orthogonal rectification.

| | | | | Real-world $\boldsymbol{A} = \boldsymbol{A}_1$ | | | | | | Synthetic $\boldsymbol{A} = \boldsymbol{A}_2$ | |
|---|---|---|---|---|---|---|---|---|---|---|---|
| Mapping $\mathcal{R}$ | Bit | QM | OR | NRE ↓ | AE (°) ↓ | | Mapping $\mathcal{R}$ | Bit | QM | OR | NRE ↓ | AE (°) ↓ |
| DT | 8 | $\boldsymbol{A}$ | ✗ | 0.2192 | 8.3014 | | DT | 8 | $\boldsymbol{A}$ | ✗ | 0.1896 | 10.877 |
| | 4 | $\boldsymbol{A}$ | ✗ | 0.6241 | 17.319 | | | 4 | $\boldsymbol{A}$ | ✗ | 0.4615 | 17.189 |
| | 4 | $\boldsymbol{U}$ | ✗ | 0.0709 | 4.0426 | | | 4 | $\boldsymbol{U}$ | ✗ | 0.1224 | 7.0144 |
| | 4 | $\boldsymbol{U}$ | ✓ | 0.0455 | 2.5615 | | | 4 | $\boldsymbol{U}$ | ✓ | 0.0878 | 4.9960 |
| Linear-2 | 8 | $\boldsymbol{A}$ | ✗ | 0.2164 | 7.9751 | | Linear-2 | 8 | $\boldsymbol{A}$ | ✗ | 0.1310 | 7.4717 |
| | 4 | $\boldsymbol{A}$ | ✗ | 0.6243 | 17.293 | | | 4 | $\boldsymbol{A}$ | ✗ | 0.4465 | 15.338 |
| | 4 | $\boldsymbol{U}$ | ✗ | 0.0543 | 3.1066 | | | 4 | $\boldsymbol{U}$ | ✗ | 0.0942 | 5.3998 |
| | 4 | $\boldsymbol{U}$ | ✓ | 0.0343 | 1.9456 | | | 4 | $\boldsymbol{U}$ | ✓ | 0.0669 | 3.8166 |

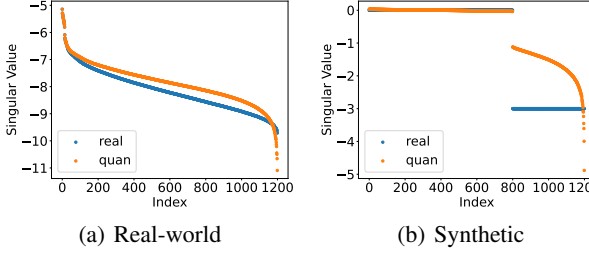
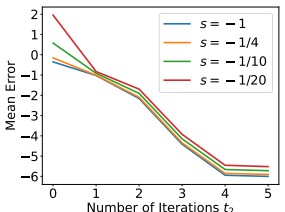

| (a) Real-world | (b) Synthetic |

Figure 2: Singular value distributions of PD matrices (real) and their 4-bit compressions (quan) used in Table 1 with $\mathcal{R}$=DT, QM=$A$. Singular values are shown on a $\log_{10}$ scale.

Figure 3: Elementwise mean errors between $(V_{t_2}\Lambda^s V_{t_2}^\mathsf{T})^{-1/s}(V_{t_2}\Lambda V_{t_2}^\mathsf{T})$ and identity matrix $I$. Mean errors are shown on a $\log_{10}$ scale.

### 3.2 Rectifying the Orthogonality of Eigenvector Matrices

Let $A$ be a PD matrix with SVD $U\Lambda U^\mathsf{T}$. Note that the eigenvector matrix $U$ is orthogonal, whereas $V = \mathcal{D}(\mathcal{Q}(U))$ may not be. To further mitigate the quantization errors mentioned in Subsection 3.1, we propose employing Björck orthonormalization [4] to orthogonalize $V$. Particularly, given $V_0 = V$, we iterate

$$V_t = 1.5V_{t-1} - 0.5V_{t-1}V_{t-1}^\mathsf{T}V_{t-1}, \tag{2}$$

for $t_1 \geq 1$ times and take $V_{t_1}$ as the rectified result. Equation (2) can also be interpreted as the gradient descent of problem $\min_V \|V^\mathsf{T}V - I\|_F^2$ using a step size of 0.5, where $I$ denotes the identity matrix. We empirically find that only one iteration (i.e., $t_1 = 1$) is enough. Table 1 illustrates the benefit of rectifying $V$ into $V_1$.

The update frequencies for the preconditioners and their inverse 4-th roots differ (see Algorithm 3). Given $V$ and $\Lambda$, we also require orthogonal rectification to compute $A^s$ rapidly for any $s \in \mathbb{R}$. The reason is as follows. It is easy to compute $A^s = U\Lambda^s U^\mathsf{T}$ by definition. However, $U\Lambda^s U^\mathsf{T}$ can be very sensitive to the orthogonality of $U$ for $s < 0$, making $V\Lambda^s V^\mathsf{T}$ largely deviate from $(V\Lambda V^\mathsf{T})^s \approx A^s$. Similarly, we can approximate $A^s$ by $V_{t_2}\Lambda^s V_{t_2}^\mathsf{T}$, where $V_{t_2}$ is generated by (2). Figure 3 illustrates the elementwise mean errors between $(V_{t_2}\Lambda^s V_{t_2}^\mathsf{T})^{-1/s}(V_{t_2}\Lambda V_{t_2}^\mathsf{T})$ and $I$ for various $s$ and $t_2$, where $A$ is the real-world matrix used in Table 1. Based on the observation from Figure 3, we set $t_2 = 4$ in our experiments.

### 3.3 Selecting the Quantizer

The quantizer $\mathcal{Q}$ is defined by the normalization operator $\mathcal{N}$ and mapping $\mathcal{R}$, and $\mathcal{N}$ is determined by $\mathbb{X}_i$. Since an eigenvector has a unit length, the elements in $\mathbb{X}_i$ should belong to the same column of an eigenvector matrix, i.e., they are from the same eigenvector. Instead of employing dynamic tree (DT) quantization as mapping $\mathcal{R}$, we recommend utilizing linear square (Linear-2) quantization as $\mathcal{R}$, particularly when $b = 4$. Linear-2 quantization is defined as

$$\mathcal{R}(j) = \begin{cases} -\left(-1 + 2j/(2^b - 1)\right)^2, & j < 2^{b-1} - 1; \\ 0, & j = 2^{b-1} - 1; \\ \left(-1 + 2j/(2^b - 1)\right)^2, & j > 2^{b-1} - 1, \end{cases} \tag{3}$$

where $j \in \mathbb{T}_b = \{0, 1, \ldots, 2^b - 1\}$. As shown in Table 1, Linear-2 quantization has lower quantization errors compared to DT quantization at 4-bit precision.

### 3.4 Overall Algorithm

We first describe the update processes of the preconditioners and their inverse 4-th roots in our 4-bit Shampoo. A preconditioner $A$ is a PD matrix and its SVD is $U\Lambda U^\mathsf{T}$. We can compress $A$ into a pair $(\lambda, \overline{U}) = (\text{diag}(\Lambda), \mathcal{Q}(U))$ and decompress it into $(\Lambda, V) = (\text{Diag}(\lambda), \mathcal{D}(\overline{U}))$. Algorithm 1 (Preconditioner Update, PU) shows the update rule of $A$. Similarly, we compress $\widehat{A} \approx A^{-1/4}$ into a pair $(a, \overline{A}) = (\text{diag}(\widehat{A}), \mathcal{Q}(\widehat{A} - \text{Diag}(a)))$ and decompress it into $\text{Diag}(a) + \mathcal{D}(\overline{A})$. Algorithm 2

(Preconditioner's Inverse 4-th Root Update, PIRU) gives the update rule of $\widehat{\boldsymbol{A}}$. Based on the above update rules, we can summarize our 4-bit Shampoo in Algorithm 3. Note that we omit some input parameters of PU and PIRU because they can be found in Algorithm 3 in the same form.

---

**Algorithm 1** PU($\boldsymbol{\lambda}, \overline{\boldsymbol{U}}, \boldsymbol{M}$)

---

**Input:** singular value vector $\boldsymbol{\lambda}$, quantized eigenvector matrix $\overline{\boldsymbol{U}}$, $\boldsymbol{M}$, number of iterations $t_1$ for rectification, exponential decay rate $\beta \in (0, 1)$, $\mathcal{Q}$ and $\mathcal{D}$
1: $\boldsymbol{\Lambda} = \mathrm{Diag}(\boldsymbol{\lambda}), \boldsymbol{V} = \mathcal{D}(\overline{\boldsymbol{U}})$
2: Rectify $\boldsymbol{V}$ by iterating (2) $t_1$ times
3: $\boldsymbol{A} = \beta \boldsymbol{V} \boldsymbol{\Lambda} \boldsymbol{V}^{\mathsf{T}} + (1-\beta)\boldsymbol{M}$
4: Compute $\boldsymbol{A} = \boldsymbol{P}\boldsymbol{\Sigma}\boldsymbol{P}^{\mathsf{T}}$ by randomized SVD
5: **return** $\mathrm{diag}(\boldsymbol{\Sigma}), \mathcal{Q}(\boldsymbol{P})$

---

**Algorithm 2** PIRU($\boldsymbol{\lambda}, \overline{\boldsymbol{U}}$)

---

**Input:** singular value vector $\boldsymbol{\lambda}$, quantized eigenvector matrix $\overline{\boldsymbol{U}}$, number of iterations $t_2$ for rectification, dampening term $\epsilon \boldsymbol{I}$, $\mathcal{Q}$ and $\mathcal{D}$
1: $\boldsymbol{\Lambda} = \mathrm{Diag}(\boldsymbol{\lambda}), \boldsymbol{V} = \mathcal{D}(\overline{\boldsymbol{U}})$
2: Rectify $\boldsymbol{V}$ by iterating (2) $t_2$ times
3: $\widehat{\boldsymbol{A}} = \boldsymbol{V}(\boldsymbol{\Lambda} + \max\{\boldsymbol{\lambda}\}\epsilon \boldsymbol{I})^{-1/4}\boldsymbol{V}^{\mathsf{T}}$
4: $\boldsymbol{a} = \mathrm{diag}(\widehat{\boldsymbol{A}})$
5: **return** $\boldsymbol{a}, \mathcal{Q}(\widehat{\boldsymbol{A}} - \mathrm{Diag}(\boldsymbol{a}))$

---

**Algorithm 3** Practical 4-bit Shampoo

---

**Input:** $\boldsymbol{W}_0 \in \mathbb{R}^{m \times n}$, $\boldsymbol{L}_0 = \epsilon \boldsymbol{I}_m$, $\boldsymbol{R}_0 = \epsilon \boldsymbol{I}_n$, $\widehat{\boldsymbol{L}}_0 = \boldsymbol{I}_m$, $\widehat{\boldsymbol{R}}_0 = \boldsymbol{I}_n$, $\beta \in (0, 1)$, $t_1, t_2$, update interval $T_1$, update interval $T_2$, total number of steps $T$, first-order optimizer $\mathcal{F}$, first-order optimizer state $\boldsymbol{s}_0 = \boldsymbol{0}$, 4-bit quantizer $\mathcal{Q}$ and its corresponding dequantizer $\mathcal{D}$.
**Output:** final parameter $\boldsymbol{W}_T$.
1: $\boldsymbol{\lambda}_{0,L} = \mathrm{diag}(\boldsymbol{L}_0), \overline{\boldsymbol{U}}_{0,L} = \mathcal{Q}(\boldsymbol{I}_m); \quad \boldsymbol{\lambda}_{0,R} = \mathrm{diag}(\boldsymbol{R}_0), \overline{\boldsymbol{U}}_{0,R} = \mathcal{Q}(\boldsymbol{I}_n)$
2: $\boldsymbol{l}_0 = \mathrm{diag}(\widehat{\boldsymbol{L}}_0), \overline{\boldsymbol{L}}_0 = \mathcal{Q}(\boldsymbol{0}); \quad \boldsymbol{r}_0 = \mathrm{diag}(\widehat{\boldsymbol{R}}_0), \overline{\boldsymbol{R}}_0 = \mathcal{Q}(\boldsymbol{0})$
3: **for** $t = 1, 2, \ldots, T$ **do**
4:     Receive loss function $\mathcal{L}_t : \mathbb{R}^{m \times n} \mapsto \mathbb{R}$ and compute gradient $\boldsymbol{G}_t = \nabla \mathcal{L}_t(\boldsymbol{W}_t)$
5:     **if** $t \% T_1 \equiv 0$ **then**
6:         $\boldsymbol{\lambda}_{t,L}, \overline{\boldsymbol{U}}_{t,L} = \mathrm{PU}(\boldsymbol{\lambda}_{t-1,L}, \overline{\boldsymbol{U}}_{t-1,L}, \boldsymbol{G}_t \boldsymbol{G}_t^{\mathsf{T}}); \; \boldsymbol{\lambda}_{t,R}, \overline{\boldsymbol{U}}_{t,R} = \mathrm{PU}(\boldsymbol{\lambda}_{t-1,R}, \overline{\boldsymbol{U}}_{t-1,R}, \boldsymbol{G}_t^{\mathsf{T}} \boldsymbol{G}_t)$
7:     **else**
8:         $\boldsymbol{\lambda}_{t,L}, \overline{\boldsymbol{U}}_{t,L} = \boldsymbol{\lambda}_{t-1,L}, \overline{\boldsymbol{U}}_{t-1,L}; \quad \boldsymbol{\lambda}_{t,R}, \overline{\boldsymbol{U}}_{t,R} = \boldsymbol{\lambda}_{t-1,R}, \overline{\boldsymbol{U}}_{t-1,R}$
9:     **if** $t \% T_2 \equiv 0$ **then**
10:      $\boldsymbol{l}_t, \overline{\boldsymbol{L}}_t = \mathrm{PIRU}(\boldsymbol{\lambda}_{t,L}, \overline{\boldsymbol{U}}_{t,L}); \quad \boldsymbol{r}_t, \overline{\boldsymbol{R}}_t = \mathrm{PIRU}(\boldsymbol{\lambda}_{t,R}, \overline{\boldsymbol{U}}_{t,R})$
11:     **else**
12:      $\boldsymbol{l}_t, \overline{\boldsymbol{L}}_t = \boldsymbol{l}_{t-1}, \overline{\boldsymbol{L}}_{t-1}; \quad \boldsymbol{r}_t, \overline{\boldsymbol{R}}_t = \boldsymbol{r}_{t-1}, \overline{\boldsymbol{R}}_{t-1}$
13:     $\widehat{\boldsymbol{L}}_t = \mathrm{Diag}(\boldsymbol{l}_t) + \mathcal{D}(\overline{\boldsymbol{L}}_t); \quad \widehat{\boldsymbol{R}}_t = \mathrm{Diag}(\boldsymbol{r}_t) + \mathcal{D}(\overline{\boldsymbol{R}}_t)$
14:     $\widehat{\boldsymbol{G}}_t = \widehat{\boldsymbol{L}}_t \boldsymbol{G}_t \widehat{\boldsymbol{R}}_t; \quad \widetilde{\boldsymbol{G}}_t = \widehat{\boldsymbol{G}}_t(\|\boldsymbol{G}_t\|_F / \|\widehat{\boldsymbol{G}}_t\|_F)$
15:     $\boldsymbol{W}_t, \boldsymbol{s}_t = \mathcal{F}(\boldsymbol{W}_{t-1}, \boldsymbol{s}_{t-1}, \widetilde{\boldsymbol{G}}_t)$

---

## 4   Theoretical Analysis

In this section, we analyze why quantizing the eigenvector matrix of a preconditioner in Shampoo is better than quantizing the preconditioner itself under a certain singular value distribution. Furthermore, we consider quantization as a perturbation and prove the convergence of the perturbed Shampoo (Algorithm 6) in Appendix E. The following lemma reveals some good properties of perturbing the eigenvector matrix of a PD matrix.

**Lemma 1.** *Let $\boldsymbol{A}$ be a PD matrix whose SVD is $\boldsymbol{U}\boldsymbol{\Lambda}\boldsymbol{U}^{\mathsf{T}}$, where $\boldsymbol{U} = [\boldsymbol{u}_i]$ is an orthogonal matrix and $\boldsymbol{\Lambda} = \mathrm{diag}([\lambda_i]^{\mathsf{T}})$ is a diagonal matrix. Given a perturbation $\Delta \boldsymbol{U} = [\Delta \boldsymbol{u}_i]$ and $s \in \mathbb{R}$, we define $\boldsymbol{B} := (\boldsymbol{U}\boldsymbol{\Lambda}\boldsymbol{U}^{\mathsf{T}})^s$ and $\Delta \boldsymbol{B} := ((\boldsymbol{U}+\Delta \boldsymbol{U})\boldsymbol{\Lambda}(\boldsymbol{U}+\Delta \boldsymbol{U})^{\mathsf{T}})^s - \boldsymbol{B}$.*

*(1) If $\boldsymbol{U}+\Delta \boldsymbol{U}$ is orthogonal and there exists $\alpha \in \mathbb{R}$ such that $\|\Delta \boldsymbol{u}_i\|_2 \leq \alpha$, then*

$$\frac{\|\Delta \boldsymbol{B}\|_F}{\|\boldsymbol{B}\|_F} \leq 2\alpha.$$

*(2) If $\boldsymbol{U}+\Delta \boldsymbol{U}$ is orthogonal and there exists $\beta \in \mathbb{R}$ such that $\langle \boldsymbol{u}_i, \boldsymbol{u}_i + \Delta \boldsymbol{u}_i \rangle \geq 1-\beta \geq 0$, then*

$$\frac{\langle \boldsymbol{B}, \boldsymbol{B}+\Delta \boldsymbol{B} \rangle}{\|\boldsymbol{B}\|_F \|\boldsymbol{B}+\Delta \boldsymbol{B}\|_F} \geq (1-\beta)^2.$$

From Lemma 1, it is evident that the normwise relative error and angle error in $f(\boldsymbol{A}) = \boldsymbol{A}^s$ of perturbing $\boldsymbol{U}$ at $\boldsymbol{A} = \boldsymbol{U}\boldsymbol{\Lambda}\boldsymbol{U}^\mathsf{T}$ are independent of $\boldsymbol{\Lambda}$ and $s$. Moreover, these errors are well-bounded under some mild conditions. Empirically, for 4-bit quantization, $\alpha = 0.1$ and $\beta = 0.005$ roughly meet the conditions of Lemma 1, leading to $\frac{\|\Delta\boldsymbol{B}\|_F}{\|\boldsymbol{B}\|_F} \leq 0.2$ and $\frac{\langle\boldsymbol{B}, \boldsymbol{B}+\Delta\boldsymbol{B}\rangle}{\|\boldsymbol{B}\|_F\|\boldsymbol{B}+\Delta\boldsymbol{B}\|_F} \geq 0.99$.

It is very complicated to generally analyze the perturbation in $f(\boldsymbol{A}) = \boldsymbol{A}^s$ of perturbing $\boldsymbol{A}$. Thus, we focus on perturbing the singular values of $\boldsymbol{A}$. For simplicity, we assume that both $\boldsymbol{A}$ and $\boldsymbol{A} + \Delta\boldsymbol{A}$ have only two distinct singular values, where $\Delta\boldsymbol{A}$ is a perturbation of $\boldsymbol{A}$. The following lemma gives the perturbation in $\boldsymbol{A}^s$ of perturbing the smaller singular value of $\boldsymbol{A}$.

**Lemma 2.** *Let $\boldsymbol{A}$ be a PD matrix of order $m+n$ whose SVD is $\boldsymbol{U}\boldsymbol{\Lambda}\boldsymbol{U}^\mathsf{T}$, where $m, n \in \mathbb{N}_+$, $n = lm$, $\boldsymbol{U} = [\boldsymbol{u}_i]$ is an orthogonal matrix and $\boldsymbol{\Lambda} = \mathrm{diag}([\lambda_i]^\mathsf{T})$ is a diagonal matrix. Assume that $\boldsymbol{\Lambda} = \mathrm{diag}([c\lambda\mathbf{1}_{m\times1}^\mathsf{T}, \lambda\mathbf{1}_{n\times1}^\mathsf{T}]^\mathsf{T})$, $c \geq 1$, and $\lambda > 0$. Given a perturbation $\Delta\boldsymbol{\Lambda} = \mathrm{diag}([\boldsymbol{0}_{m\times1}^\mathsf{T}, \Delta\boldsymbol{\lambda}_{n\times1}^\mathsf{T}]^\mathsf{T})$ and $s \in \mathbb{R}$, we define $\boldsymbol{B} := (\boldsymbol{U}\boldsymbol{\Lambda}\boldsymbol{U}^\mathsf{T})^s$ and $\Delta\boldsymbol{B} := (\boldsymbol{U}(\boldsymbol{\Lambda}+\Delta\boldsymbol{\Lambda})\boldsymbol{U}^\mathsf{T})^s - \boldsymbol{B}$.*

*(1) If $\Delta\boldsymbol{\lambda}_{n\times1} = (k-1)\lambda\mathbf{1}_{n\times1}$ where $k > 0$, then*

$$\frac{\|\Delta\boldsymbol{B}\|_F}{\|\boldsymbol{B}\|_F} = \frac{\sqrt{l}|k^s - 1|}{\sqrt{c^{2s} + l}} = h_1(s, l).$$

*Moreover, $h_1(s, l)$ decreases monotonically with $s$ over $(-\infty, 0)$ and increases monotonically with $l$ over $(0, +\infty)$.*

*(2) If $\Delta\boldsymbol{\lambda}_{n\times1} = (tc-1)\lambda\mathbf{1}_{n\times1}$ where $t > 0$, then*

$$\frac{\langle\boldsymbol{B}, \boldsymbol{B}+\Delta\boldsymbol{B}\rangle}{\|\boldsymbol{B}\|_F\|\boldsymbol{B}+\Delta\boldsymbol{B}\|_F} = \frac{lt^s + c^s}{\sqrt{(1 + lt^{2s})(l + c^{2s})}} = h_2(l).$$

*Moreover, $h_2(l)$ decreases monotonically with $l$ over $(0, (c/t)^s]$ and increases monotonically with $l$ over $((c/t)^s, +\infty)$.*

*(3) If $\Delta\boldsymbol{\lambda}_{n\times1} = (tc-1)\lambda\mathbf{1}_{n\times1}$ where $k = tc > 0$ and $l = (c/t)^s$, then*

$$\frac{\|\Delta\boldsymbol{B}\|_F}{\|\boldsymbol{B}\|_F} = \frac{|k^s - 1|}{\sqrt{k^s + 1}}, \quad \frac{\langle\boldsymbol{B}, \boldsymbol{B}+\Delta\boldsymbol{B}\rangle}{\|\boldsymbol{B}\|_F\|\boldsymbol{B}+\Delta\boldsymbol{B}\|_F} = \frac{2}{\sqrt{2 + k^s + 1/k^s}}.$$

Let us make some comments on the above lemma. First, from Lemma 2(1) we have $h_1(1, l) = \frac{\|\Delta\boldsymbol{A}\|_F}{\|\boldsymbol{A}\|_F} = \frac{\sqrt{l}|k-1|}{\sqrt{c^2+l}}$. If $k \geq 1$, $\frac{\|\Delta\boldsymbol{A}\|_F}{\|\boldsymbol{A}\|_F} = \frac{\|\Delta\boldsymbol{\Lambda}\|_F}{\|\boldsymbol{\Lambda}\|_F}$ is bounded by $\frac{k}{c}\sqrt{l} = t\sqrt{l}$. Second, if $k = tc \geq 1$ and $s < 0$, one can deduce $h_2(l) \geq \sqrt{lt^{2s}/(1 + lt^{2s})}$ from Lemma 2(2), which indicates that a small $lt^{2s}$ is needed to achieve small $h_2(l)$. We can set $t = 0.02$ to simulate 4-bit quantization. Based on Lemma 1 and Lemma 2(3), we have the following proposition.

**Proposition 1.** *Let $\boldsymbol{A}$ be a PD matrix of order $m+n$ whose SVD is $\boldsymbol{U}\boldsymbol{\Lambda}\boldsymbol{U}^\mathsf{T}$, where $m, n \in \mathbb{N}_+$, $n = lm$, $\boldsymbol{U} = [\boldsymbol{u}_i]$ is an orthogonal matrix, $\boldsymbol{\Lambda} = \mathrm{diag}([c\lambda\mathbf{1}_{m\times1}^\mathsf{T}, \lambda\mathbf{1}_{n\times1}^\mathsf{T}]^\mathsf{T})$, $c \geq 1000$, and $\lambda > 0$. Given $\Delta\boldsymbol{U} = [\Delta\boldsymbol{u}_i]$, $\Delta\boldsymbol{\Lambda} = \mathrm{diag}([\boldsymbol{0}_{m\times1}^\mathsf{T}, \Delta\boldsymbol{\lambda}_{n\times1}^\mathsf{T}]^\mathsf{T})$, and $s \leq -0.25$, we define $\boldsymbol{B} := (\boldsymbol{U}\boldsymbol{\Lambda}\boldsymbol{U}^\mathsf{T})^s$, $\boldsymbol{B}_1 := ((\boldsymbol{U}+\Delta\boldsymbol{U})\boldsymbol{\Lambda}(\boldsymbol{U}+\Delta\boldsymbol{U})^\mathsf{T})^s$, and $\boldsymbol{B}_2 := (\boldsymbol{U}(\boldsymbol{\Lambda}+\Delta\boldsymbol{\Lambda})\boldsymbol{U}^\mathsf{T})^s$. If $\boldsymbol{U} + \Delta\boldsymbol{U}$ is orthogonal, $\|\Delta\boldsymbol{u}_i\|_2 \leq 0.1$, $\langle\boldsymbol{u}_i, \Delta\boldsymbol{u}_i\rangle \geq -0.005$, $\Delta\boldsymbol{\lambda}_{n\times1} = (0.02c-1)\lambda\mathbf{1}_{n\times1}$, and $l = (c/0.02)^s$, then*

$$2\frac{\|\boldsymbol{B}_1 - \boldsymbol{B}\|_F}{\|\boldsymbol{B}\|_F} \leq 0.4 \leq \frac{\|\boldsymbol{B}_2 - \boldsymbol{B}\|_F}{\|\boldsymbol{B}\|_F}, \quad 6\left(1 - \frac{\langle\boldsymbol{B}, \boldsymbol{B}_1\rangle}{\|\boldsymbol{B}\|_F\|\boldsymbol{B}_1\|_F}\right) \leq 0.06 \leq \left(1 - \frac{\langle\boldsymbol{B}, \boldsymbol{B}_2\rangle}{\|\boldsymbol{B}\|_F\|\boldsymbol{B}_2\|_F}\right).$$

Proposition 1 requires very strong assumptions. Nevertheless, it provides insight into why quantizing $\boldsymbol{A}$ can result in a greater normwise relative error and angle error in $\boldsymbol{A}^s$, compared to quantizing $\boldsymbol{U}$. Complete proofs of Lemma 1, Lemma 2, and Proposition 1 can be found in Appendix F.

## 5 Experiments

In this section, we compare our 4-bit Shampoo combined with SGDM or AdamW to their 32-bit counterparts, as well as the first-order optimizers on various image classification tasks. See more experimental results on image classification and natural language modeling tasks in Appendix H.

**Models, datasets, and hyperparameters.** We train VGG19 [36], ResNet34 [20], ViT-Small [10], and Swin-Tiny [28] on the CIFAR-100 [23] and Tiny-ImageNet [24] datasets with one RTX3060Ti GPU, and train ResNet50 and ViT-Base/32 on the ImageNet-1k dataset [34] with one A800 GPU.

Table 2: Performance, wall-clock time and memory cost on various image classification tasks. TA = test accuracy, WCT = wall-clock time, and TMC = total GPU memory cost.

| Dataset | Model | Optimizer | TA (%) | WCT (min) | TMC (MB) |
|---|---|---|---|---|---|
| CIFAR-100 | VGG19 | SGDM | 74.14 | 97.70 | 512.17 |
| | | SGDM + 32-bit Shampoo | 74.54 | 84.45 | 979.13 |
| | | SGDM + 4-bit Shampoo | 74.74 | 92.51 | 577.14 |
| | ResNet34 | SGDM | 78.98 | 170.1 | 822.03 |
| | | SGDM + 32-bit Shampoo | 79.71 | 147.2 | 1441.8 |
| | | SGDM + 4-bit Shampoo | 79.17 | 155.8 | 908.40 |
| | ViT-Small | AdamW | 74.34 | 668.1 | 2720.0 |
| | | AdamW + 32-bit Shampoo | 77.50 | 498.7 | 3252.0 |
| | | AdamW + 4-bit Shampoo | 77.22 | 510.8 | 2791.7 |
| | Swin-Tiny | AdamW | 76.69 | 318.6 | 1465.8 |
| | | AdamW + 32-bit Shampoo | 79.34 | 260.8 | 2036.0 |
| | | AdamW + 4-bit Shampoo | 78.63 | 273.3 | 1543.9 |
| Tiny-ImageNet | VGG19 | SGDM | 61.53 | 172.0 | 1062.3 |
| | | SGDM + 32-bit Shampoo | 63.39 | 136.5 | 1531.9 |
| | | SGDM + 4-bit Shampoo | 62.84 | 143.8 | 1127.3 |
| | ResNet34 | SGDM | 67.10 | 432.1 | 2304.0 |
| | | SGDM + 32-bit Shampoo | 67.90 | 313.0 | 2924.3 |
| | | SGDM + 4-bit Shampoo | 67.95 | 329.3 | 2390.4 |
| | ViT-Small | AdamW | 54.66 | 1274 | 2730.1 |
| | | AdamW + 32-bit Shampoo | 57.11 | 953.9 | 3261.1 |
| | | AdamW + 4-bit Shampoo | 57.15 | 970.3 | 2801.9 |
| | Swin-Tiny | AdamW | 58.77 | 701.9 | 1789.9 |
| | | AdamW + 32-bit Shampoo | 61.74 | 565.3 | 2362.8 |
| | | AdamW + 4-bit Shampoo | 62.24 | 582.7 | 1868.1 |
| ImageNet-1k | ResNet50 | SGDM | 76.70 | 2134 | 11307 |
| | | SGDM + 32-bit Shampoo | 77.07 | 1910 | 11937 |
| | | SGDM + 4-bit Shampoo | 76.92 | 1970 | 11396 |
| | ViT-Base/32 | AdamW | 72.87 | 2190 | 10600 |
| | | AdamW + 32-bit Shampoo | 75.03 | 1774 | 12134 |
| | | AdamW + 4-bit Shampoo | 74.78 | 1770 | 10804 |

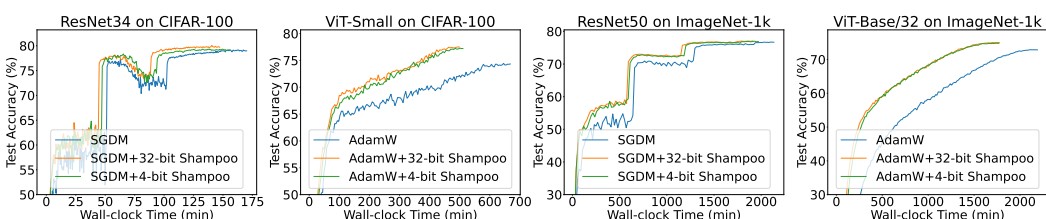

Figure 4: Visualization of test accuracies on the CIFAR-100 and ImageNet-1k datasets.

For hyperparameter settings, we mainly follow [41] to train CNNs and [25, 44] to train vision transformers. For all the tasks, we keep the common hyperparameters of optimizers the same values. See Appendix G for experimental details.

**Main results.** We show the performance, wall-clock time, and memory cost in Table 2. First-order optimizers run 1.2x to 1.5x epochs, resulting in longer wall-clock time, yet yielding lower test accuracies compared to second-order optimizers. In comparison to 32-bit Shampoo, our 4-bit Shampoo shows comparable test accuracies with differences ranging from -0.7% to 0.5%, increases in wall-clock time varying from -0.2% to 9.5%, and memory savings of 4.5% to 41%. Compared to the first-order optimizers, the memory costs of our 4-bit Shampoo only rise by 0.8% to 12.7%. This represents a significant advancement in the utilization of second-order optimizers. Following [26], we report the total peak GPU memory consumption rather than the optimizer's peak GPU memory consumption. Our main focus is on quantizing the states for constructing preconditioners and their inverse roots, which are approximately 7x smaller for 4-bit Shampoo compared to 32-bit Shampoo (see Appendix G). Figure 4 shows the test accuracy curves on the CIFAR-100 and ImageNet-1k

Table 3: Ablation study on the impact of different quantization techniques to Swin-Tiny training on the CIFAR-100 dataset. $U$ is the eigenvector matrix of a preconditioner $A$. QM = quantized matrix, OR = orthogonal rectification in Algorithm 1, TL = training loss, and TA = test accuracy.

| 4-bit | | | | | 3-bit | | | | |
|---|---|---|---|---|---|---|---|---|---|
| Mapping $\mathcal{R}$ | QM | OR | TL | TA (%) | Mapping $\mathcal{R}$ | QM | OR | TL | TA (%) |
| Linear-2 | $A$ | ✗ | 1.631 | 76.95 | Linear-2 | $A$ | ✗ | 1.648 | 76.70 |
| DT | $U$ | ✗ | 1.569 | 78.70 | DT | $U$ | ✗ | NaN | - |
| Linear-2 | $U$ | ✗ | 1.566 | 78.22 | Linear-2 | $U$ | ✗ | NaN | - |
| Linear-2 | $U$ | ✓ | 1.551 | 78.63 | Linear-2 | $U$ | ✓ | 1.572 | 78.53 |

datasets. The test accuracy curves of 4-bit Shampoo and 32-bit Shampoo are very close, both of which are above the test accuracy curves of the first-order optimizers.

**Ablations.** We investigate the effectiveness of our proposed quantization techniques. Table 3 indicates that quantizing the eigenvector matrix of a preconditioner is crucial for $b$-bit ($b = 3, 4$) Shampoo to maintain 32-bit performance, and orthogonal rectification is highly beneficial for 3-bit Shampoo. As for quantization mapping, linear square (Linear-2) quantization is comparable to dynamic tree (DT) quantization. We further apply our 4-bit quantization techniques to K-FAC [30], AdaBK [41] and CASPR [13] and the results are shown in Table 4. We can see that the 4-bit optimizers match the performance of their 32-bit counterparts, and reduce memory by over 20%.

## 6   Related Work

**Second-order optimizers.** Different second-order optimizers apply different second-order information. Hessian-based optimizers [39, 27] use the Hessian matrix or its approximation. Fisher-based optimizers [30, 41] utilize the covariance matrix of the accumulated gradients or its approximation based on Kronecker product. Shampoo [18] and CASPR [13] approximate the full AdaGrad [12] preconditioner by a set of small preconditioning matrices.

Table 4: Performance and memory cost of training Swin-Tiny on CIFAR-100. TA = test accuracy and TMC = total GPU memory cost.

| Optimizer | TA (%) | TMC (MB) |
|---|---|---|
| AdamW+32-bit K-FAC | 78.20 | 2388.0 |
| AdamW+4-bit K-FAC | 78.56 | 1878.3 |
| 32-bit AdamW_BK | 79.28 | 2388.0 |
| 4-bit AdamW_BK | 79.34 | 1878.3 |
| AdamW+32-bit CASPR | 78.82 | 2034.6 |
| AdamW+4-bit CASPR | 78.80 | 1543.9 |

**Memory efficient optimizers based on factorization.** Adafactor [35] employs the outer product of two vectors to approximate the second moment of Adam [22]. SM3 [3] considers approximating the second moment of Adam by its covers' statistics. [14] and [40] reduce memory cost of the preconditioner in a second-order optimizer with its low-rank approximation through truncated SVD.

**Memory efficient optimizers based on quantization.** Dettmers et al. [8] introduce block-wise dynamic quantization that enables the use of first-order optimizers with 8-bit states. Li et al. [26] push the optimizer states of Adam/AdamW to 4-bit.

## 7   Conclusions, Limitations, and Broader Impact

We propose 4-bit Shampoo, the first low-bit second-order optimizer, designed for memory-efficient training of DNNs. We find that quantizing the eigenvector matrix of the preconditioner is essential to minimize quantization errors in its inverse 4-th root at 4-bit precision, given its sensitivity to alterations in small singular values. We further introduce orthogonal rectification and linear square quantization mapping to improve performance. 4-bit Shampoo achieves lossless performance to 32-bit counterpart in training different DNNs on various tasks.

**Limitations.** Preconditioners in Shampoo are symmetric matrices and can be stored as upper triangular matrices, saving almost half of the memory usage. However, the eigenvector matrix of a preconditioner is not symmetric, causing an 8-bit preconditioner to occupy the same memory as its 4-bit eigenvector matrix. Notably, a comparison of Table 1 and Table 7 in Appendix D shows that the 4-bit quantization of the eigenvector matrix has smaller quantization errors than the 8-bit quantization of the preconditioner. Our evaluation is currently limited to image classification and natural language modeling tasks. Due to limitations in computing resources, we do not test our 4-bit Shampoo on large-scale models with billions of parameters.

**Broader Impact.** Our work can facilitate training large models with second-order optimizers. This could open up new research possibilities that were previously unattainable due to GPU memory constraints, especially benefiting researchers with limited resources.

## Acknowledgments and Disclosure of Funding

Jia Li and Hua Huang were supported by the NSF of China (grant no. 62131003). Jia Li was also supported by the NSF of China (grant no. 62102034). Pan Zhou was supported by the Singapore Ministry of Education (MOE) Academic Research Fund (AcRF) Tier 1 grants (project ID: 23-SIS-SMU-028 and 23-SIS-SMU-070).

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

# A Implementation Details of Shampoo, CASPR, K-FAC and AdaBK

The implementation of 32-bit Shampoo used in our experiments is described in Algorithm 4. Our Pytorch implementation of Shampoo is partially based on the code provided by [2]. We implement CASPR by replacing $\widehat{G}_t = \widehat{L}_t G_t \widehat{R}_t$ with $J_t = \widehat{L}_t G_t + G_t \widehat{R}_t; \widehat{G}_t = \widehat{L}_t J_t + J_t \widehat{R}_t$ in line 12 of Algorithm 4 and line 14 of Algorithm 3. We summarize the implementation of 32-bit K-FAC/AdaBK in Algorithm 5, where $X_t$ is the input feature and $Y_t$ is the output feature gradient. Both power iteration [5] and Schur-Newton iteration [17] are run for 10 iterations. Our implementation of 4-bit K-FAC/AdaBK is similar to 4-bit Shampoo (i.e., compressing $L_t, R_t, \widehat{L}_t$, and $\widehat{R}_t$).

---

**Algorithm 4** Practical 32-bit Shampoo

---

**Input:** initial parameter $W_0 \in \mathbb{R}^{m \times n}$, left preconditioner $L_0 = \epsilon I_m$, right preconditioner $R_0 = \epsilon I_n$, inverse root of left preconditioner $\widehat{L}_0 = I_m$, inverse root of right preconditioner $\widehat{R}_0 = I_n$, total number of steps $T$, interval of updating preconditioners $T_1$, interval of updating inverse roots of preconditioners $T_2$, exponential decay rate for preconditioners $\beta \in (0, 1)$, first-order optimizer $\mathcal{F}$, first-order optimizer state $s_0 = 0$.

**Output:** final parameter $W_T$.

1: **for** $t = 1, 2, \ldots, T$ **do**
2:     Receive loss function $\mathcal{L}_t : \mathbb{R}^{m \times n} \mapsto \mathbb{R}$ and compute gradient $G_t = \nabla \mathcal{L}_t(W_t)$
3:     **if** $t\%T_1 \equiv 0$ **then**
4:         $L_t = \beta L_{t-1} + (1 - \beta) G_t G_t^\mathsf{T}; \quad R_t = \beta R_{t-1} + (1 - \beta) G_t^\mathsf{T} G_t$
5:     **else**
6:         $L_t = L_{t-1}; \quad R_t = R_{t-1}$
7:     **if** $t\%T_2 \equiv 0$ **then**
8:         Compute maximum eigenvalues $\lambda_{\max}^L$ and $\lambda_{\max}^R$ of $L_t$ and $R_t$ by power iteration
9:         Compute $\widehat{L}_t = (L_t + \lambda_{\max}^L \epsilon I_m)^{-1/4}$ and $\widehat{R}_t = (R_t + \lambda_{\max}^R \epsilon I_n)^{-1/4}$ by Schur-Newton iteration
10:     **else**
11:         $\widehat{L}_t = \widehat{L}_{t-1}; \quad \widehat{R}_t = \widehat{R}_{t-1}$
12:     $\widehat{G}_t = \widehat{L}_t G_t \widehat{R}_t; \quad \widetilde{G}_t = \widehat{G}_t(\|G_t\|_F / \|\widehat{G}_t\|_F)$
13:     $W_t, s_t = \mathcal{F}(W_{t-1}, s_{t-1}, \widetilde{G}_t)$

---

**Algorithm 5** Practical 32-bit K-FAC/AdaBK

---

**Input:** initial parameter $W_0 \in \mathbb{R}^{m \times n}$, left preconditioner $L_0 = 0$, right preconditioner $R_0 = 0$, inverse root of left preconditioner $\widehat{L}_0 = I_m$, inverse root of right preconditioner $\widehat{R}_0 = I_n$, total number of steps $T$, interval of updating preconditioners $T_1$, interval of updating inverse roots of preconditioners $T_2$, $\epsilon$, exponential decay rate for preconditioners $\beta \in (0, 1)$, $\alpha = 1$ for K-FAC / $\alpha = 2$ for AdaBK, first-order optimizer $\mathcal{F}$, first-order optimizer state $s_0 = 0$.

**Output:** final parameter $W_T$.

1: **for** $t = 1, 2, \ldots, T$ **do**
2:     Receive loss function $\mathcal{L}_t : \mathbb{R}^{m \times n} \mapsto \mathbb{R}$ and compute gradient $G_t = \nabla \mathcal{L}_t(W_t)$
3:     Receive $X_t$ by forward propagation and $Y_t$ by backward propagation
4:     **if** $t\%T_1 \equiv 0$ **then**
5:         $L_t = \beta L_{t-1} + (1 - \beta) Y_t Y_t^\mathsf{T}; \quad R_t = \beta R_{t-1} + (1 - \beta) X_t X_t^\mathsf{T}$
6:     **else**
7:         $L_t = L_{t-1}; \quad R_t = R_{t-1}$
8:     **if** $t\%T_2 \equiv 0$ **then**
9:         Compute maximum eigenvalues $\lambda_{\max}^L$ and $\lambda_{\max}^R$ of $L_t$ and $R_t$ by power iteration
10:         Compute $\widehat{L}_t = (L_t + \lambda_{\max}^L \epsilon I_m)^{-1/\alpha}$ and $\widehat{R}_t = (R_t + \lambda_{\max}^R \epsilon I_n)^{-1/\alpha}$ by Schur-Newton iteration
11:     **else**
12:         $\widehat{L}_t = \widehat{L}_{t-1}; \quad \widehat{R}_t = \widehat{R}_{t-1}$
13:     $\widehat{G}_t = \widehat{L}_t G_t \widehat{R}_t; \quad \widetilde{G}_t = \widehat{G}_t(\|G_t\|_F / \|\widehat{G}_t\|_F)$
14:     $W_t, s_t = \mathcal{F}(W_{t-1}, s_{t-1}, \widetilde{G}_t)$

---

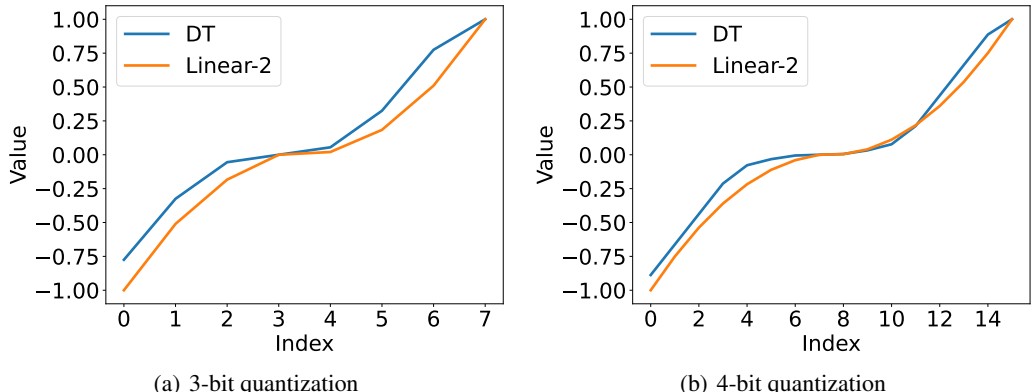

(a) 3-bit quantization          (b) 4-bit quantization

Figure 5: Visualization of DT quantization and Linear-2 quantization at $b$-bit ($b = 3, 4$) precision.

## B    Randomized SVD Method

Given an initial matrix $\boldsymbol{P}_0 \in \mathbb{R}^{n \times n}$, randomized SVD method computes the eigenvector matrix of a PD matrix $\boldsymbol{A} \in \mathbb{R}^{n \times n}$ by iterating

$$\boldsymbol{P}_t = \mathrm{QR}(\boldsymbol{A}\boldsymbol{P}_{t-1}), \tag{4}$$

where $\mathrm{QR}(\boldsymbol{X})$ denotes the QR decomposition of matrix $\boldsymbol{X}$, returning an orthogonal matrix. Since we can initialize $\boldsymbol{P}_0$ with the previous result (e.g., $\boldsymbol{V}$ in Algorithm 1), only a few iterations are enough to obtain an accurate estimation in practice. In our experiments, we iterate (4) once for Shampoo/CASPR, and iterate (4) twice for K-FAC/AdaBK.

## C    Quantization Mappings

We present the constructions of different quantization mappings in $b$-bit quantizers ($\mathcal{R}$ in $\mathcal{Q}$). See Figure 5 for the illustration of them. Note that $\mathbb{T}_b = \{0, 1, \ldots, 2^b - 1\}$.

Dynamic tree (DT) quantization for $b$-bit quantization maps $\mathbb{T}_b$ onto $\{0, 1\} \cup G$, where $G$ is a set of numbers with the following properties: the number in $G$ looks like $\pm q_k \times 10^{-E}$, where a) $b = 2 + E + F$, where $E, F$ are integers; b) $q_k = (p_k + p_{k+1})/2$, where $k \in \{0, \ldots, 2^F - 1\}$; c) $p_j = 0.9j/2^F + 0.1$, where $j \in \{0, \ldots, 2^F\}$. For 4-bit quantization, DT quantization maps $\mathbb{T}_4$ onto {-0.8875, -0.6625, -0.4375, -0.2125, -0.0775, -0.0325, -0.0055, 0.0000, 0.0055, 0.0325, 0.0775, 0.2125, 0.4375, 0.6625, 0.8875, 1.0000}. For 3-bit quantization, DT quantization maps $\mathbb{T}_3$ onto {-0.7750, -0.3250, -0.0550, 0.0000, 0.0550, 0.3250, 0.7750, 1.0000}.

For 4-bit quantization, linear square (Linear-2) quantization maps $\mathbb{T}_4$ onto {-1.0000, -0.7511, -0.5378, -0.3600, -0.2178, -0.1111, -0.0400, 0.0000, 0.0044, 0.0400, 0.1111, 0.2178, 0.3600, 0.5378, 0.7511, 1.0000}. For 3-bit quantization, Linear-2 quantization maps $\mathbb{T}_3$ onto {-1.0000, -0.5102, -0.1837, 0.0000, 0.0204, 0.1837, 0.5102, 1.0000}.

## D    Quantization Error Analyses

We present more quantization error analyses of the preconditioners. Recall that we define two kinds of quantization errors in mapping $f$ of transformation $g$ at $\boldsymbol{A} \in \mathbb{R}^{m \times n}$ (in short errors in $f(\boldsymbol{A})$ of $g$) in Subsection 3.1. Here we extend them as follows: define the normwise relative error (NRE) in $f$ of $(g_1, g_2)$ at $\boldsymbol{A}$ as

$$\mathrm{NRE} = \frac{\|f(\boldsymbol{A}) - g_2 \circ f \circ g_1(\boldsymbol{A})\|_F}{\|f(\boldsymbol{A})\|_F},$$

and the angle error (AE) in $f$ of $(g_1, g_2)$ at $\boldsymbol{A}$ as

$$\mathrm{AE} = \arccos\left(\frac{\langle f(\boldsymbol{A}), g_2 \circ f \circ g_1(\boldsymbol{A})\rangle}{\|f(\boldsymbol{A})\|_F \|g_2 \circ f \circ g_1(\boldsymbol{A})\|_F}\right).$$

## D.1 Static Analysis

Table 5 is an extension of Table 1 for Bit=4. Since the diagonal elements of $\boldsymbol{A}^{-1/4}$ are usually much larger than its non-diagonal elements where $\boldsymbol{A}$ is a PD matrix, we further consider the quantization errors in $f(\boldsymbol{A}) = \boldsymbol{A}^{-1/4} - \mathrm{Diag}(\mathrm{diag}(\boldsymbol{A}^{-1/4}))$ at 4-bit precision as shown in Table 6. Table 7 shows the quantization errors at 8-bit precision.

A large condition number of a PD matrix $\boldsymbol{A}$ is indispensable for the superiority of quantizing $\boldsymbol{U}$ over quantizing $\boldsymbol{A}$, where $\boldsymbol{U}$ is the eigenvector matrix of $\boldsymbol{A}$. We consider contracting the singular value distribution of $\boldsymbol{A} = \boldsymbol{A}_1$ with SVD $\boldsymbol{U}\mathrm{Diag}(\boldsymbol{\lambda})\boldsymbol{U}^\mathsf{T}$ used in Table 5 by mapping each singular value $\lambda$ of $\boldsymbol{A}$ to $h(\lambda) = \tau(\lambda - \lambda_{\min}^A) + \lambda_{\min}^A$, where $\lambda_{\min}^A$ is the minimum singular value of $\boldsymbol{A}$ and $\tau > 0$ is the contraction coefficient. Figure 6 shows 4-bit quantization errors in $\boldsymbol{A}^{-1/4}$ or $\boldsymbol{A}^{-1/4} - \mathrm{Diag}(\mathrm{diag}(\boldsymbol{A}^{-1/4}))$ of quantizing $\boldsymbol{U}$ or $\boldsymbol{A}$ at $\boldsymbol{A} = \boldsymbol{U}\mathrm{Diag}(h(\boldsymbol{\lambda}))\boldsymbol{U}^\mathsf{T}$.

Table 5: Quantization errors in $f(\boldsymbol{A}) = \boldsymbol{A}^{-1/4}$ of different 4-bit quantization schemes at a PD matrix $\boldsymbol{A}$. We employ block-wise normalization with a block size of 64. $\boldsymbol{U}$ is the eigenvector matrix of $\boldsymbol{A}$ and $\boldsymbol{B} = (g_1(\boldsymbol{A}))^{-1/4}$. QM = quantized matrices and OR = orthogonal rectification.

| | | | Real-world $\boldsymbol{A} = \boldsymbol{A}_1$ | | | Synthetic $\boldsymbol{A} = \boldsymbol{A}_2$ | | |
|---|---|---|---|---|---|---|---|---|
| Mapping $\mathcal{R}$ | QM | OR | NRE $\downarrow$ | AE (°) $\downarrow$ | Mapping $\mathcal{R}$ | QM | OR | NRE $\downarrow$ | AE (°) $\downarrow$ |
| DT | $\boldsymbol{A}$ | ✗ | 0.6241 | 17.319 | DT | $\boldsymbol{A}$ | ✗ | 0.4615 | 17.189 |
| | $\boldsymbol{U}$ | ✗ | 0.0709 | 4.0426 | | $\boldsymbol{U}$ | ✗ | 0.1224 | 7.0144 |
| | $\boldsymbol{U}$ | ✓ | 0.0455 | 2.5615 | | $\boldsymbol{U}$ | ✓ | 0.0878 | 4.9960 |
| | $\boldsymbol{B}$ | ✗ | 0.0398 | 2.2802 | | $\boldsymbol{B}$ | ✗ | 0.0853 | 4.8914 |
| | $(\boldsymbol{A}, \boldsymbol{B})$ | ✗ | 0.6243 | 17.364 | | $(\boldsymbol{A}, \boldsymbol{B})$ | ✗ | 0.4649 | 17.650 |
| | $(\boldsymbol{U}, \boldsymbol{B})$ | ✗ | 0.0811 | 4.6296 | | $(\boldsymbol{U}, \boldsymbol{B})$ | ✗ | 0.1485 | 8.5168 |
| | $(\boldsymbol{U}, \boldsymbol{B})$ | ✓ | 0.0604 | 3.4230 | | $(\boldsymbol{U}, \boldsymbol{B})$ | ✓ | 0.1224 | 6.9817 |
| Linear-2 | $\boldsymbol{A}$ | ✗ | 0.6243 | 17.293 | Linear-2 | $\boldsymbol{A}$ | ✗ | 0.4465 | 15.338 |
| | $\boldsymbol{U}$ | ✗ | 0.0543 | 3.1066 | | $\boldsymbol{U}$ | ✗ | 0.0942 | 5.3998 |
| | $\boldsymbol{U}$ | ✓ | 0.0343 | 1.9456 | | $\boldsymbol{U}$ | ✓ | 0.0669 | 3.8166 |
| | $\boldsymbol{B}$ | ✗ | 0.0315 | 1.8050 | | $\boldsymbol{B}$ | ✗ | 0.0661 | 3.7887 |
| | $(\boldsymbol{A}, \boldsymbol{B})$ | ✗ | 0.6243 | 17.301 | | $(\boldsymbol{A}, \boldsymbol{B})$ | ✗ | 0.4483 | 15.654 |
| | $(\boldsymbol{U}, \boldsymbol{B})$ | ✗ | 0.0626 | 3.5833 | | $(\boldsymbol{U}, \boldsymbol{B})$ | ✗ | 0.1150 | 6.5901 |
| | $(\boldsymbol{U}, \boldsymbol{B})$ | ✓ | 0.0466 | 2.6494 | | $(\boldsymbol{U}, \boldsymbol{B})$ | ✓ | 0.0941 | 5.3716 |

Table 6: Quantization errors in $f(\boldsymbol{A}) = \boldsymbol{A}^{-1/4} - \mathrm{Diag}(\mathrm{diag}(\boldsymbol{A}^{-1/4}))$ of different 4-bit quantization schemes at a PD matrix $\boldsymbol{A}$. We employ block-wise normalization with a block size of 64. $\boldsymbol{U}$ is the eigenvector matrix of $\boldsymbol{A}$ and $\boldsymbol{B} = (g_1(\boldsymbol{A}))^{-1/4}$. QM = quantized matrices and OR = orthogonal rectification.

| | | | Real-world $\boldsymbol{A} = \boldsymbol{A}_1$ | | | Synthetic $\boldsymbol{A} = \boldsymbol{A}_2$ | | |
|---|---|---|---|---|---|---|---|---|
| Mapping $\mathcal{R}$ | QM | OR | NRE $\downarrow$ | AE (°) $\downarrow$ | Mapping $\mathcal{R}$ | QM | OR | NRE $\downarrow$ | AE (°) $\downarrow$ |
| DT | $\boldsymbol{A}$ | ✗ | 0.9549 | 59.360 | DT | $\boldsymbol{A}$ | ✗ | 0.6247 | 25.913 |
| | $\boldsymbol{U}$ | ✗ | 0.2328 | 13.287 | | $\boldsymbol{U}$ | ✗ | 0.1994 | 11.444 |
| | $\boldsymbol{U}$ | ✓ | 0.1480 | 8.4365 | | $\boldsymbol{U}$ | ✓ | 0.1427 | 8.1415 |
| | $\boldsymbol{B}$ | ✗ | 0.1314 | 7.5513 | | $\boldsymbol{B}$ | ✗ | 0.1391 | 7.9813 |
| | $(\boldsymbol{A}, \boldsymbol{B})$ | ✗ | 0.9561 | 59.825 | | $(\boldsymbol{A}, \boldsymbol{B})$ | ✗ | 0.6314 | 26.948 |
| | $(\boldsymbol{U}, \boldsymbol{B})$ | ✗ | 0.2666 | 15.281 | | $(\boldsymbol{U}, \boldsymbol{B})$ | ✗ | 0.2420 | 13.911 |
| | $(\boldsymbol{U}, \boldsymbol{B})$ | ✓ | 0.1977 | 11.322 | | $(\boldsymbol{U}, \boldsymbol{B})$ | ✓ | 0.1992 | 11.393 |
| Linear-2 | $\boldsymbol{A}$ | ✗ | 0.9547 | 58.336 | Linear-2 | $\boldsymbol{A}$ | ✗ | 0.6010 | 20.780 |
| | $\boldsymbol{U}$ | ✗ | 0.1786 | 10.213 | | $\boldsymbol{U}$ | ✗ | 0.1534 | 8.8027 |
| | $\boldsymbol{U}$ | ✓ | 0.1122 | 6.4096 | | $\boldsymbol{U}$ | ✓ | 0.1088 | 6.2176 |
| | $\boldsymbol{B}$ | ✗ | 0.1041 | 5.9554 | | $\boldsymbol{B}$ | ✗ | 0.1078 | 6.1755 |
| | $(\boldsymbol{A}, \boldsymbol{B})$ | ✗ | 0.9548 | 58.601 | | $(\boldsymbol{A}, \boldsymbol{B})$ | ✗ | 0.6047 | 21.666 |
| | $(\boldsymbol{U}, \boldsymbol{B})$ | ✗ | 0.2063 | 11.778 | | $(\boldsymbol{U}, \boldsymbol{B})$ | ✗ | 0.1873 | 10.745 |
| | $(\boldsymbol{U}, \boldsymbol{B})$ | ✓ | 0.1530 | 8.7337 | | $(\boldsymbol{U}, \boldsymbol{B})$ | ✓ | 0.1532 | 8.7534 |

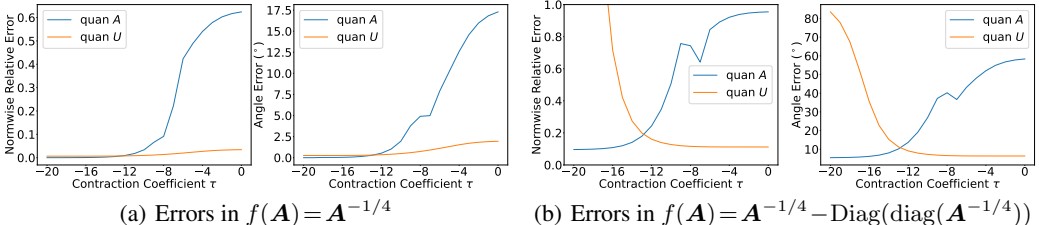

(a) Errors in $f(\boldsymbol{A}) = \boldsymbol{A}^{-1/4}$        (b) Errors in $f(\boldsymbol{A}) = \boldsymbol{A}^{-1/4} - \mathrm{Diag}(\mathrm{diag}(\boldsymbol{A}^{-1/4}))$

Figure 6: 4-bit quantization errors in $f(\boldsymbol{A})$ of quantizing $\boldsymbol{U}$ or $\boldsymbol{A}$ at $\boldsymbol{A} = \boldsymbol{U}\mathrm{Diag}(h(\boldsymbol{\lambda}))\boldsymbol{U}^{\mathsf{T}}$. We use linear square quantization and orthogonal rectification. The condition number $\mathrm{cond}(\boldsymbol{A}) = \lambda_{\max}^{A}/\lambda_{\min}^{A}$ is around 37235, where $\lambda_{\max}^{A}$ and $\lambda_{\min}^{A}$ are the maximum and minimum singular values of $\boldsymbol{A}$ respectively. Contraction coefficients are shown on a $\log_2$ scale.

Table 7: Quantization errors in $f(\boldsymbol{A})$ of different 8-bit quantization schemes at a PD matrix $\boldsymbol{A}$, where $\boldsymbol{A} = \boldsymbol{A}_1$ is derived from the real world as described in Subsection 3.1. We employ block-wise normalization with a block size of 256. $\boldsymbol{U}$ is the eigenvector matrix of $\boldsymbol{A}$ and $\boldsymbol{B} = (g_1(\boldsymbol{A}))^{-1/4}$. QM = quantized matrices and OR = orthogonal rectification.

| | | | $f(\boldsymbol{A}) = \boldsymbol{A}^{-1/4}$ | | | | | $f(\boldsymbol{A}) = \boldsymbol{A}^{-1/4} - \mathrm{Diag}(\mathrm{diag}(\boldsymbol{A}^{-1/4}))$ | |
|---|---|---|---|---|---|---|---|---|---|
| Mapping $\mathcal{R}$ | QM | OR | NRE ↓ | AE (°) ↓ | Mapping $\mathcal{R}$ | QM | OR | NRE ↓ | AE (°) ↓ |
| DT | $\boldsymbol{A}$ | ✗ | 0.2192 | 8.3014 | DT | $\boldsymbol{A}$ | ✗ | 0.5001 | 23.644 |
| | $\boldsymbol{U}$ | ✗ | 0.0060 | 0.3421 | | $\boldsymbol{U}$ | ✗ | 0.0197 | 1.1273 |
| | $\boldsymbol{U}$ | ✓ | 0.0037 | 0.2140 | | $\boldsymbol{U}$ | ✓ | 0.0123 | 0.7022 |
| | $\boldsymbol{B}$ | ✗ | 0.0029 | 0.1655 | | $\boldsymbol{B}$ | ✗ | 0.0097 | 0.5553 |
| | $(\boldsymbol{A}, \boldsymbol{B})$ | ✗ | 0.2193 | 8.3051 | | $(\boldsymbol{A}, \boldsymbol{B})$ | ✗ | 0.5003 | 23.649 |
| | $(\boldsymbol{U}, \boldsymbol{B})$ | ✗ | 0.0067 | 0.3810 | | $(\boldsymbol{U}, \boldsymbol{B})$ | ✗ | 0.0219 | 1.2577 |
| | $(\boldsymbol{U}, \boldsymbol{B})$ | ✓ | 0.0047 | 0.2712 | | $(\boldsymbol{U}, \boldsymbol{B})$ | ✓ | 0.0156 | 0.8955 |
| Linear-2 | $\boldsymbol{A}$ | ✗ | 0.2164 | 7.9751 | Linear-2 | $\boldsymbol{A}$ | ✗ | 0.4875 | 21.447 |
| | $\boldsymbol{U}$ | ✗ | 0.0037 | 0.2121 | | $\boldsymbol{U}$ | ✗ | 0.0122 | 0.6994 |
| | $\boldsymbol{U}$ | ✓ | 0.0023 | 0.1312 | | $\boldsymbol{U}$ | ✓ | 0.0076 | 0.4343 |
| | $\boldsymbol{B}$ | ✗ | 0.0021 | 0.1203 | | $\boldsymbol{B}$ | ✗ | 0.0070 | 0.4035 |
| | $(\boldsymbol{A}, \boldsymbol{B})$ | ✗ | 0.2164 | 7.9755 | | $(\boldsymbol{A}, \boldsymbol{B})$ | ✗ | 0.4875 | 21.448 |
| | $(\boldsymbol{U}, \boldsymbol{B})$ | ✗ | 0.0043 | 0.2439 | | $(\boldsymbol{U}, \boldsymbol{B})$ | ✗ | 0.0141 | 0.8079 |
| | $(\boldsymbol{U}, \boldsymbol{B})$ | ✓ | 0.0031 | 0.1791 | | $(\boldsymbol{U}, \boldsymbol{B})$ | ✓ | 0.0104 | 0.5935 |

### D.2 Dynamic Analysis

We define the normwise relative error (NRE) and angle error (AE) of $\boldsymbol{B}$ deviating from $\boldsymbol{A}$ as

$$\mathrm{NRE} = \frac{\|\boldsymbol{B} - \boldsymbol{A}\|_F}{\|\boldsymbol{A}\|_F}, \quad \mathrm{AE} = \arccos\left(\frac{\langle \boldsymbol{A}, \boldsymbol{B}\rangle}{\|\boldsymbol{A}\|_F\|\boldsymbol{B}\|_F}\right).$$

Consider Shampoo using 4-bit preconditioners for parameter updates, but also recording 32-bit preconditioners at the same time. We extract the left preconditioners $\boldsymbol{L}_4$ and $\boldsymbol{L}_{32} \in \mathbb{R}^{1200 \times 1200}$ of a specific model parameter block $\boldsymbol{W} \in \mathbb{R}^{1200 \times 768}$ every 8000 steps in the Swin-Tiny training on CIFAR-100 with AdamW+Shampoo. Here $\boldsymbol{L}_4$ is a decompressed 4-bit preconditioner, and $\boldsymbol{L}_{32}$ is a 32-bit preconditioner.

Figure 7 shows the quantization errors during training. For naive 4-bit Shampoo, $\boldsymbol{L}_{32}^{-1/4}$ and $\boldsymbol{L}_4^{-1/4}$ are computed by Schur-Newton iteration used in Algorithm 4 where $\epsilon = 10^{-4}$. For our 4-bit Shampoo, $\boldsymbol{L}_{32}^{-1/4}$ is computed by Schur-Newton iteration used in Algorithm 4 where $\epsilon = 10^{-4}$, and $\boldsymbol{L}_4^{-1/4}$ is computed by Algorithm 2 without quantization where $\epsilon = 10^{-4}, t_2 = 4$. We find that $\epsilon = 10^{-6}$ for Algorithm 2 used in our main experiments though is effective, yet it can cause a large numerical instability in the later stage of training (see Figure 8).

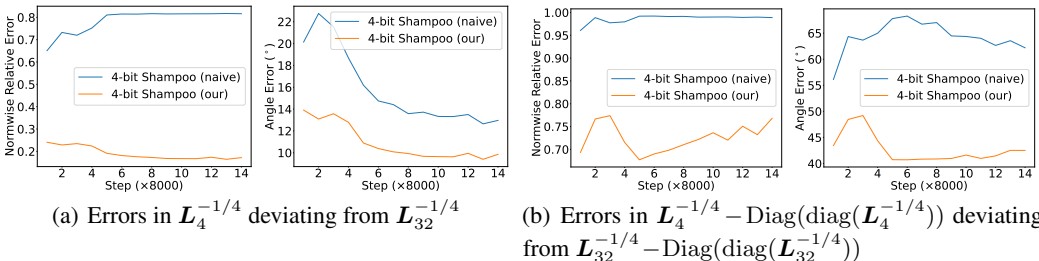

(a) Errors in $\boldsymbol{L}_4^{-1/4}$ deviating from $\boldsymbol{L}_{32}^{-1/4}$
(b) Errors in $\boldsymbol{L}_4^{-1/4} - \mathrm{Diag}(\mathrm{diag}(\boldsymbol{L}_4^{-1/4}))$ deviating from $\boldsymbol{L}_{32}^{-1/4} - \mathrm{Diag}(\mathrm{diag}(\boldsymbol{L}_{32}^{-1/4}))$

Figure 7: Quantization errors during Swin-Tiny training on the CIFAR-100 dataset. We use dampening term $\epsilon = 10^{-4}$ to compute $\boldsymbol{L}_4^{-1/4}$ and $\boldsymbol{L}_{32}^{-1/4}$.

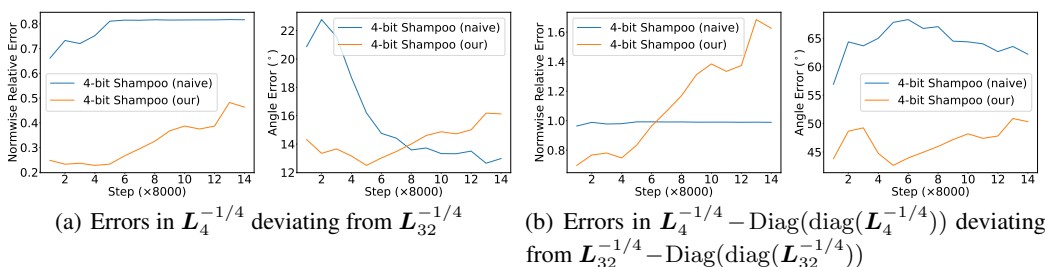

(a) Errors in $\boldsymbol{L}_4^{-1/4}$ deviating from $\boldsymbol{L}_{32}^{-1/4}$
(b) Errors in $\boldsymbol{L}_4^{-1/4} - \mathrm{Diag}(\mathrm{diag}(\boldsymbol{L}_4^{-1/4}))$ deviating from $\boldsymbol{L}_{32}^{-1/4} - \mathrm{Diag}(\mathrm{diag}(\boldsymbol{L}_{32}^{-1/4}))$

Figure 8: Quantization errors during Swin-Tiny training on the CIFAR-100 dataset. We use dampening term $\epsilon = 10^{-6}$ to compute $\boldsymbol{L}_4^{-1/4}$ and $\boldsymbol{L}_{32}^{-1/4}$.

## E  Convergence Analysis

**More notations.** Given a symmetric real matrix $\boldsymbol{A}$, $\boldsymbol{A} \succeq 0$ means that $\boldsymbol{A}$ is positive semidefinite (PSD), and $\boldsymbol{A} \succ 0$ means that $\boldsymbol{A}$ is positive definite (PD). Assume that symmetric matrices $\boldsymbol{A}$ and $\boldsymbol{B}$ are symmetric, the notations $\boldsymbol{A} \succeq \boldsymbol{B}$ and $\boldsymbol{A} \succ \boldsymbol{B}$ mean that $\boldsymbol{A} - \boldsymbol{B} \succeq 0$ and $\boldsymbol{A} - \boldsymbol{B} \succ 0$ respectively. Let $\boldsymbol{A}$ be a PSD matrix and $s \in \mathbb{R}$, we define $\boldsymbol{A}^s = \boldsymbol{U}\boldsymbol{\Lambda}^s\boldsymbol{U}^\mathsf{T}$, where $\boldsymbol{U}\boldsymbol{\Lambda}\boldsymbol{U}^\mathsf{T}$ is the Singular Value Decomposition (SVD) of $\boldsymbol{A}$. The Mahalanobis norm of a vector $\boldsymbol{x}$ induced by a PD matrix $\boldsymbol{A}$ is $\|\boldsymbol{x}\|_{\boldsymbol{A}} = \sqrt{\boldsymbol{x}^\mathsf{T}\boldsymbol{A}\boldsymbol{x}}$. The dual norm of $\|\cdot\|_{\boldsymbol{A}}$ is denoted by $\|\cdot\|_{\boldsymbol{A}}^*$, where $\|\boldsymbol{x}\|_{\boldsymbol{A}}^* = \sqrt{\boldsymbol{x}^\mathsf{T}\boldsymbol{A}^{-1}\boldsymbol{x}}$. The spectral norm of matrix $\boldsymbol{A}$ is $\|\boldsymbol{A}\|_2 = \sup_{\boldsymbol{x} \neq \boldsymbol{0}}\{\|\boldsymbol{A}\boldsymbol{x}\|_2/\|\boldsymbol{x}\|_2\}$. $\boldsymbol{A} \otimes \boldsymbol{B}$ means the (right) Kronecker product of matrices $\boldsymbol{A}$ and $\boldsymbol{B}$. $\overline{\mathrm{vec}}(\boldsymbol{A})$ means the vectorization (stacking the rows) of $\boldsymbol{A}$.

---

**Algorithm 6** Perturbed Shampoo in the matrix case

---

**Input:** $\boldsymbol{W}_0 \in \mathbb{R}^{m \times n}, \boldsymbol{L}_0 = \boldsymbol{0}_{m \times m}, \boldsymbol{R}_0 = \boldsymbol{0}_{n \times n}, \rho_0 = 0, \mu_0 = 0.$
1: **for** $t = 1, \ldots, T$ **do**
2:   Receive loss function: $f_t : \mathbb{R}^{m \times n} \to \mathbb{R}$
3:   Compute gradient: $\boldsymbol{G}_t = \nabla f_t(\boldsymbol{W}_t)$
4:   Update preconditioners: $\boldsymbol{J}_t = \boldsymbol{L}_{t-1} + \boldsymbol{G}_t\boldsymbol{G}_t^\mathsf{T}; \quad \boldsymbol{K}_t = \boldsymbol{R}_{t-1} + \boldsymbol{G}_t^\mathsf{T}\boldsymbol{G}_t$
5:   Perturb preconditioners: $\boldsymbol{L}_t = g(\boldsymbol{J}_t); \quad \boldsymbol{R}_t = g(\boldsymbol{K}_t)$
6:   Accumulate errors: $\rho_t = \rho_{t-1} + \|\boldsymbol{J}_t - \boldsymbol{L}_t\|_2; \quad \mu_t = \mu_{t-1} + \|\boldsymbol{K}_t - \boldsymbol{R}_t\|_2$
7:   Update parameters: $\boldsymbol{W}_{t+1} = \boldsymbol{W}_t - \eta((\epsilon + \rho_t)\boldsymbol{I}_m + \boldsymbol{L}_t)^{-1/4}\boldsymbol{G}_t((\epsilon + \mu_t)\boldsymbol{I}_n + \boldsymbol{R}_t)^{-1/4}$

---

We consider quantization as a perturbation and present the perturbed Shampoo in Algorithm 6 for convergence analysis. The regret bound of the perturbed Shampoo can be found in Theorem 1. Complete proofs can be found in Appendix F. We first introduce some basic technical tools, and the details of them are in [18, 21].

**Lemma 3.** *Let $\boldsymbol{A}, \boldsymbol{A}', \boldsymbol{B}, \boldsymbol{B}'$ be matrices of appropriate dimensions, and $\boldsymbol{u}, \boldsymbol{v}$ be two column vectors. The following properties hold:*

*(1) $(\boldsymbol{A} \otimes \boldsymbol{B})(\boldsymbol{A}' \otimes \boldsymbol{B}') = (\boldsymbol{A}\boldsymbol{A}') \otimes (\boldsymbol{B}\boldsymbol{B}')$;*

(2) $(\boldsymbol{A} \otimes \boldsymbol{B})^{\mathsf{T}} = (\boldsymbol{A}^{\mathsf{T}} \otimes \boldsymbol{B}^{\mathsf{T}})$;

(3) If $\boldsymbol{A}, \boldsymbol{B} \succeq 0$ and $s \in \mathbb{R}$, then $(\boldsymbol{A} \otimes \boldsymbol{B})^s = (\boldsymbol{A}^s \otimes \boldsymbol{B}^s)$;

(4) If $\boldsymbol{A} \succeq \boldsymbol{A}'$ and $\boldsymbol{B} \succeq \boldsymbol{B}'$, then $\boldsymbol{A} \otimes \boldsymbol{B} \succeq \boldsymbol{A}' \otimes \boldsymbol{B}'$;

(5) $\operatorname{tr}(\boldsymbol{A}\bar{\boldsymbol{B}}) = \operatorname{tr}(\boldsymbol{A})\operatorname{tr}(\boldsymbol{B})$;

(6) $\overline{\operatorname{vec}}(\boldsymbol{u}\boldsymbol{v}^{\mathsf{T}}) = \boldsymbol{u} \otimes \boldsymbol{v}$.

**Lemma 4.** *Let $\boldsymbol{G} \in \mathbb{R}^{m \times n}, \boldsymbol{L} \in \mathbb{R}^{m \times m}, \boldsymbol{R} \in \mathbb{R}^{n \times n}$, then it holds that*
$$(\boldsymbol{L} \otimes \boldsymbol{R}^{\mathsf{T}})\overline{\operatorname{vec}}(\boldsymbol{G}) = \overline{\operatorname{vec}}(\boldsymbol{LGR}).$$

**Lemma 5.** *Assume that $0 \preceq \boldsymbol{X}_i \preceq \boldsymbol{Y}_i$ for $i = 1, \ldots, n$. Assume further that all $\boldsymbol{X}_i$ commute with each other and all $\boldsymbol{Y}_i$ commute with each other. Let $\alpha_1, \ldots, \alpha_n \geq 0$ such that $\sum_{i=1}^n \alpha_i = 1$, then*
$$\boldsymbol{X}_1^{\alpha_1} \cdots \boldsymbol{X}_n^{\alpha_n} \preceq \boldsymbol{Y}_1^{\alpha_1} \cdots \boldsymbol{Y}_n^{\alpha_n}.$$

**Lemma 6.** *Let $0 \leq \alpha \leq 1$ and $0 \preceq \boldsymbol{X} \preceq \boldsymbol{Y}$, then $\boldsymbol{X}^\alpha \preceq \boldsymbol{Y}^\alpha$.*

**Lemma 7.** *Let $\boldsymbol{A} \succ 0$ and $\boldsymbol{B} \succ 0$, then it holds that $\boldsymbol{A} \succeq \boldsymbol{B}$ if and only if $\boldsymbol{B}^{-1} \succeq \boldsymbol{A}^{-1}$.*

**Lemma 8** (von Neumann). *Let $\boldsymbol{A}, \boldsymbol{B} \in \mathbb{R}^{m \times n}$ and $q = \min\{m, n\}$. Let $\sigma_1(\boldsymbol{A}) \geq \cdots \geq \sigma_q(\boldsymbol{A})$ and $\sigma_1(\boldsymbol{B}) \geq \cdots \geq \sigma_q(\boldsymbol{B})$ denote the non-increasingly ordered singular values of $\boldsymbol{A}$ and $\boldsymbol{B}$, respectively. Then*
$$\langle \boldsymbol{A}, \boldsymbol{B} \rangle \leq \sum_{i=1}^q \sigma_i(\boldsymbol{A})\sigma_i(\boldsymbol{B}).$$

**Lemma 9.** *Assume that function $f_t$ is continuously differentiable and convex on $\mathbb{R}^d$, and matrix $\boldsymbol{H}_t \succ 0$ for $t = 1, \ldots, T$. Given $\boldsymbol{w}_0 \in \mathbb{R}^d, \eta > 0$, define $\boldsymbol{w}_{t+1} = \boldsymbol{w}_t - \eta \boldsymbol{H}_t^{-1} \boldsymbol{g}_t$, where $\boldsymbol{g}_t = \nabla f_t(\boldsymbol{w}_t)$. Then for any $\boldsymbol{w}^* \in \mathbb{R}^d$, we have*
$$\sum_{t=1}^T f_t(\boldsymbol{w}_t) - \sum_{t=1}^T f_t(\boldsymbol{w}^*) \leq \frac{1}{2\eta} \sum_{t=1}^T (\|\boldsymbol{w}_t - \boldsymbol{w}^*\|_{\boldsymbol{H}_t}^2 - \|\boldsymbol{w}_{t+1} - \boldsymbol{w}^*\|_{\boldsymbol{H}_t}^2) + \frac{\eta}{2} \sum_{t=1}^T (\|\boldsymbol{g}_t\|_{\boldsymbol{H}_t}^*)^2.$$

**Lemma 10.** *Let $\boldsymbol{g}_1, \ldots, \boldsymbol{g}_T$ be a sequence of vectors. For $\rho > 0$, define $\widehat{\boldsymbol{H}}_t = (\rho \boldsymbol{I} + \sum_{s=1}^t \boldsymbol{g}_s \boldsymbol{g}_s^{\mathsf{T}})^{1/2}$. Then we have*
$$\sum_{t=1}^T (\|\boldsymbol{g}_t\|_{\widehat{\boldsymbol{H}}_t}^*)^2 \leq 2\operatorname{tr}(\widehat{\boldsymbol{H}}_T).$$

**Lemma 11.** *Assume that $\boldsymbol{G}_1, \ldots, \boldsymbol{G}_T \in \mathbb{R}^{m \times n}$ are matrices of rank at most $r$. Let $s$ for $t = 1, \ldots, T$. Then for any $\epsilon \geq 0$,*
$$\epsilon \boldsymbol{I}_{mn} + \frac{1}{r} \sum_{t=1}^T \boldsymbol{g}_t \boldsymbol{g}_t^{\mathsf{T}} \preceq (\epsilon \boldsymbol{I}_m + \sum_{t=1}^T \boldsymbol{G}_t \boldsymbol{G}_t^{\mathsf{T}})^{1/2} \otimes (\epsilon \boldsymbol{I}_n + \sum_{t=1}^T \boldsymbol{G}_t^{\mathsf{T}} \boldsymbol{G}_t)^{1/2}.$$

The key to the convergence proof of Algorithm 6 is forming a PD matrix sequence $\{\boldsymbol{H}_i\}_{i=1}^T$, which satisfies $0 \prec \boldsymbol{H}_1 \preceq \cdots \preceq \boldsymbol{H}_T$. To achieve it, we gives the following lemma extended from Lemma 2 in the Appendix of [40].

**Lemma 12.** *Let $\{\boldsymbol{X}_t\}_{t=1}^{t=T}$ be a sequence of symmetric matrices, and $\boldsymbol{A}_t = \sum_{s=1}^t \boldsymbol{X}_s$, where $t = 1, \ldots, T$. Suppose we have two sequences of symmetric matrices $\{\boldsymbol{Y}_t\}_{t=1}^{t=T}, \{\boldsymbol{Z}_t\}_{t=0}^{t=T}$, and a sequence real numbers $\{\rho_t\}_{t=0}^{t=T}$ satisfying*
$$\boldsymbol{Y}_t = \boldsymbol{Z}_{t-1} + \boldsymbol{X}_t, \quad \rho_t = \rho_{t-1} + \|\boldsymbol{Y}_t - \boldsymbol{Z}_t\|_2, \quad \boldsymbol{Z}_0 = \boldsymbol{0}, \rho_0 = 0.$$
*Define $\boldsymbol{B}_t = \rho_t \boldsymbol{I} + \boldsymbol{Z}_t$, where $\boldsymbol{I}$ denotes the identity matrix. Then for $t = 1, \ldots, T$, we have*
$$\boldsymbol{B}_t \succeq \boldsymbol{B}_{t-1} + \boldsymbol{X}_t, \quad \boldsymbol{A}_t \preceq \boldsymbol{B}_t \preceq 2\rho_t \boldsymbol{I} + \boldsymbol{A}_t.$$

**Theorem 1.** *Assume that the gradients $\boldsymbol{G}_1, \ldots, \boldsymbol{G}_T \in \mathbb{R}^{m \times n}$ are matrices of rank at most $r$. Then for any $\boldsymbol{W}^* \in \mathbb{R}^{m \times n}$ and $\epsilon > 0$, if $\eta = D/\sqrt{2r}$, the regret of Algorithm 6 is bounded as follows,*
$$\sum_{t=1}^T f_t(\boldsymbol{W}_t) - \sum_{t=1}^T f_t(\boldsymbol{W}^*) \leq \sqrt{2r}D[2^{1/4}m\rho_T^{1/4} + \operatorname{tr}(\tilde{\boldsymbol{L}}_T^{1/4})][2^{1/4}n\mu_T^{1/4} + \operatorname{tr}(\tilde{\boldsymbol{R}}_T^{1/4})],$$
*where $D = \max_{t \in [T]} \|\boldsymbol{W}_t - \boldsymbol{W}^*\|_F$, $\tilde{\boldsymbol{L}}_t = \epsilon \boldsymbol{I}_m + \sum_{t=1}^T \boldsymbol{G}_t \boldsymbol{G}_t^{\mathsf{T}}$, and $\tilde{\boldsymbol{R}}_t = \epsilon \boldsymbol{I}_n + \sum_{t=1}^T \boldsymbol{G}_t^{\mathsf{T}} \boldsymbol{G}_t$.*

Though we get a convergence guarantee of Algorithm 6, the upper bound given by Theorem 1 is very slack, since $2^{1/4}m\rho_T^{1/4}$ is about the same as $\operatorname{tr}(\tilde{\boldsymbol{L}}_T^{1/4})$ for 4-bit quantization schemes in practice.

## F Proofs

**Lemma 1.** *Let $A$ be a PD matrix whose SVD is $U\Lambda U^\mathsf{T}$, where $U = [u_i]$ is an orthogonal matrix and $\Lambda = \mathrm{diag}([\lambda_i]^\mathsf{T})$ is a diagonal matrix. Given a perturbation $\Delta U = [\Delta u_i]$ and $s \in \mathbb{R}$, we define $B := (U\Lambda U^\mathsf{T})^s$ and $\Delta B := ((U + \Delta U)\Lambda(U + \Delta U)^\mathsf{T})^s - B$.*

*(1) If $U + \Delta U$ is orthogonal and there exists $\alpha \in \mathbb{R}$ such that $\|\Delta u_i\|_2 \leq \alpha$, then*

$$\frac{\|\Delta B\|_F}{\|B\|_F} \leq 2\alpha.$$

*(2) If $U + \Delta U$ is orthogonal and there exists $\beta \in \mathbb{R}$ such that $\langle u_i, u_i + \Delta u_i \rangle \geq 1 - \beta \geq 0$, then*

$$\frac{\langle B, B + \Delta B \rangle}{\|B\|_F \|B + \Delta B\|_F} \geq (1 - \beta)^2.$$

*Proof.* (1) Since $U$ and $U + \Delta U$ are orthogonal, we have

$$B = U\Lambda^s U^\mathsf{T}, \quad B + \Delta B = (U + \Delta U)\Lambda^s(U + \Delta U)^\mathsf{T},$$

by definition. This leads to

$$\Delta B = U\Lambda^s \Delta U^\mathsf{T} + \Delta U\Lambda^s(U + \Delta U)^\mathsf{T}.$$

The Frobenius norm satisfies the triangle inequality and is orthogonality invariant. Hence,

$$
\begin{aligned}
\|\Delta B\|_F &= \|U\Lambda^s \Delta U^\mathsf{T} + \Delta U\Lambda^s(U + \Delta U)^\mathsf{T}\|_F \\
&\leq \|U\Lambda^s \Delta U^\mathsf{T}\|_F + \|\Delta U\Lambda^s(U + \Delta U)^\mathsf{T}\|_F \\
&= \|\Lambda^s \Delta U^\mathsf{T}\|_F + \|\Delta U\Lambda^s\|_F = 2\|\Delta U\Lambda^s\|_F \\
&= 2\sqrt{\sum_i \|\lambda_i^s \Delta u_i\|_2^2} = 2\sqrt{\sum_i \lambda_i^{2s}\|\Delta u_i\|_2^2} \\
&\leq 2\sqrt{\sum_i \lambda_i^{2s}\alpha^2} = 2\alpha\sqrt{\sum_i \lambda_i^{2s}} = 2\alpha\|\Lambda^s\|_F \\
&= 2\alpha\|B\|_F.
\end{aligned}
$$

(2) Similar to (1), we have

$$\Delta B = U\Lambda^s \Delta U^\mathsf{T} + \Delta U\Lambda^s U^\mathsf{T} + \Delta U\Lambda^s \Delta U^\mathsf{T}.$$

From $\langle u_i, u_i + \Delta u_i \rangle \geq 1 - \beta \geq 0$, we get $0 \geq \langle u_i, \Delta u_i \rangle \geq -\beta \geq -1$ because

$$1 = \|u_i\|_2\|u_i + \Delta u_i\|_2 \geq \langle u_i, u_i + \Delta u_i \rangle = 1 + \langle u_i, \Delta u_i \rangle \geq 1 - \beta \geq 0,$$

holds due to the orthogonality of $U$ and $U + \Delta U$. Hence,

$$
\begin{aligned}
\langle B, \Delta B \rangle &= \mathrm{tr}(2U\Lambda^{2s}\Delta U^\mathsf{T} + U\Lambda^s U^\mathsf{T}\Delta U\Lambda^s \Delta U^\mathsf{T}) \\
&= \mathrm{tr}\Big(\sum_i 2\lambda_i^{2s} u_i \Delta u_i^\mathsf{T}\Big) + \mathrm{tr}\Big[\Big(\sum_i \lambda_i^s u_i u_i^\mathsf{T}\Big)\Big(\sum_j \lambda_j^s \Delta u_j \Delta u_j^\mathsf{T}\Big)\Big] \\
&= \Big(\sum_i 2\lambda_i^{2s}\langle u_i, \Delta u_i \rangle\Big) + \Big(\sum_{ij}\lambda_i^s \lambda_j^s \langle u_i, \Delta u_j \rangle^2\Big) \\
&\geq \Big(\sum_i 2\lambda_i^{2s}\langle u_i, \Delta u_i \rangle\Big) + \Big(\sum_i \lambda_i^{2s}\langle u_i, \Delta u_i \rangle^2\Big) \\
&= \sum_i \lambda_i^{2s}[(1 + \langle u_i, \Delta u_i \rangle)^2 - 1] \\
&\geq \sum_i \lambda_i^{2s}[(1 - \beta)^2 - 1] = [(1 - \beta)^2 - 1]\|\Lambda^s\|_F^2 \\
&= [(1 - \beta)^2 - 1]\|B\|_F^2 = [(1 - \beta)^2 - 1]\langle B, B \rangle.
\end{aligned}
$$

Therefore, we have

$$\frac{\langle B, B + \Delta B \rangle}{\|B\|_F\|B + \Delta B\|_F} = \frac{\langle B, B + \Delta B \rangle}{\langle B, B \rangle} = 1 + \frac{\langle B, \Delta B \rangle}{\langle B, B \rangle} \geq (1 - \beta)^2.$$

The proof is completed. $\square$

**Lemma 2.** *Let $A$ be a PD matrix of order $m+n$ whose SVD is $U\Lambda U^\mathsf{T}$, where $m, n \in \mathbb{N}_+$, $n = lm$, $U = [u_i]$ is an orthogonal matrix and $\Lambda = \mathrm{diag}([\lambda_i]^\mathsf{T})$ is a diagonal matrix. Assume that $\Lambda = \mathrm{diag}([c\lambda\mathbf{1}_{m\times1}^\mathsf{T}, \lambda\mathbf{1}_{n\times1}^\mathsf{T}])$, $c \geq 1$, and $\lambda > 0$. Given a perturbation $\Delta\Lambda = \mathrm{diag}([\mathbf{0}_{m\times1}^\mathsf{T}, \Delta\lambda_{n\times1}^\mathsf{T}]^\mathsf{T})$ and $s \in \mathbb{R}$, we define $B := (U\Lambda U^\mathsf{T})^s$ and $\Delta B := (U(\Lambda+\Delta\Lambda)U^\mathsf{T})^s - B$.*

*(1) If $\Delta\lambda_{n\times1} = (k-1)\lambda\mathbf{1}_{n\times1}$ where $k > 0$, then*

$$\frac{\|\Delta B\|_F}{\|B\|_F} = \frac{\sqrt{l}|k^s - 1|}{\sqrt{c^{2s} + l}} = h_1(s, l).$$

*Moreover, $h_1(s, l)$ decreases monotonically with $s$ over $(-\infty, 0)$ and increases monotonically with $l$ over $(0, +\infty)$.*

*(2) If $\Delta\lambda_{n\times1} = (tc-1)\lambda\mathbf{1}_{n\times1}$ where $t > 0$, then*

$$\frac{\langle B, B+\Delta B\rangle}{\|B\|_F\|B+\Delta B\|_F} = \frac{lt^s + c^s}{\sqrt{(1+lt^{2s})(l+c^{2s})}} = h_2(l).$$

*Moreover, $h_2(l)$ decreases monotonically with $l$ over $(0, (c/t)^s]$ and increases monotonically with $l$ over $((c/t)^s, +\infty)$.*

*(3) If $\Delta\lambda_{n\times1} = (tc-1)\lambda\mathbf{1}_{n\times1}$ where $k = tc > 0$ and $l = (c/t)^s$, then*

$$\frac{\|\Delta B\|_F}{\|B\|_F} = \frac{|k^s - 1|}{\sqrt{k^s + 1}}, \quad \frac{\langle B, B+\Delta B\rangle}{\|B\|_F\|B+\Delta B\|_F} = \frac{2}{\sqrt{2 + k^s + 1/k^s}}.$$

*Proof.* (1) Since $U$ is orthogonal, we have

$$\|\Delta B\|_F = \|(\Lambda+\Delta\Lambda)^s - \Lambda^s\|_F = \sqrt{n}|k^s - 1|\lambda^s, \quad \|B\|_F = \|\Lambda^s\|_F = \sqrt{mc^{2s} + n}\lambda^s.$$

Hence,

$$\frac{\|\Delta B\|_F}{\|B\|_F} = \frac{\sqrt{n}|k^s - 1|}{\sqrt{mc^{2s} + n}} = \frac{\sqrt{l}|k^s - 1|}{\sqrt{c^{2s} + l}} = h_1(s, l) \geq 0.$$

It is easy to check that $h_1$ increases monotonically with $l$ over $(0, +\infty)$. To prove $h_1$ decreases monotonically with $s$ over $(-\infty, 0)$, define

$$g_1(s) = \frac{1}{l}(h_1(s, l))^2 = \frac{(k^s - 1)^2}{c^{2s} + l}.$$

Consider the derivative of $g_1$

$$\begin{aligned}
g_1'(s) &= \frac{(c^{2s} + l)2(k^s - 1)k^s \ln k - (k^s - 1)^2 c^{2s}2\ln c}{(c^{2s} + l)^2} \\
&= \frac{2(k^s - 1)\left((c^{2s} + l)k^s \ln k - (k^s - 1)c^{2s}\ln c\right)}{(c^{2s} + l)^2}.
\end{aligned}$$

If $s < 0$ and $k > 1$, then $k^s - 1 < 0$, $k^s \ln k > 0$ leading to $g_1'(s) < 0$ since $c \geq 0$; Similarly, if $s < 0$ and $0 < k \leq 1$, then $k^s - 1 \geq 0$, $k^s \ln k \leq 0$ leading to $g_1'(s) \leq 0$. Thus $g_1(s)$ is a monotonically decreasing function for $s < 0$, which implies that $h_1$ decreases monotonically with $s$ over $(-\infty, 0)$.

(2) Similar to (1), we have

$$\|B\|_F = \sqrt{mc^{2s} + n}\lambda^s, \quad \|B+\Delta B\|_F = \sqrt{nt^{2s} + mc^s}\lambda^s.$$

Besides,

$$\langle B, B+\Delta B\rangle = \mathrm{tr}(U\Lambda^s(\Lambda+\Delta\Lambda)^s U^\mathsf{T}) = \mathrm{tr}(\Lambda^s(\Lambda+\Delta\Lambda)^s) = (mc^{2s} + nc^st^s)\lambda^{2s}.$$

Hence, we get

$$\frac{\langle B, B+\Delta B\rangle}{\|B\|_F\|B+\Delta B\|_F} = \frac{nt^s + mc^s}{\sqrt{(m + nt^{2s})(n + mc^{2s})}} = \frac{lt^s + c^s}{\sqrt{(1+lt^{2s})(l+c^{2s})}} = h_2(l) \geq 0.$$

To prove $h_2$ decreases monotonically with $l$ over $(0, (c/t)^s]$ and increases monotonically with $l$ over $((c/t)^s, +\infty)$, we define

$$g_2(l) = (h_2(l))^2 = \frac{(lt^s + c^s)^2}{(1 + lt^{2s})(l + c^{2s})},$$

whose monotonicity is equivalent to that of $h_2$ for $l > 0$. Consider the derivative of $g_2$

$$g_2'(l) = \left( \frac{t^{2s}l^2 + 2t^s c^s l + c^{2s}}{t^{2s}l^2 + l + t^{2s}c^{2s}l + c^{2s}} \right)' = \frac{(t^s - t^{2s}c^s)^2 l^2 - (c^s - t^s c^{2s})^2}{(t^{2s}l^2 + l + t^{2s}c^{2s}l + c^{2s})^2}.$$

If $s = 0$ or $tc = 1$, then $g_2(l) \equiv 1$. If $s \neq 0$ and $tc \neq 1$, then $(t^s - t^{2s}c^s)^2 > 0, (c^s - t^s c^{2s})^2 > 0$. In this case, let $g_2'(l) = 0$, we get

$$t^{2s}(1 - t^s c^s)^2 l^2 = c^{2s}(1 - t^s c^s)^2,$$

which implies that $l = (c/t)^s$. It is easy to see that $g_2$ decreases monotonically with $l$ over $(0, (c/t)^s]$ and increases monotonically with $l$ over $((c/t)^s, +\infty)$.

(3) According to (1)(2), we can easily get

$$\frac{\|\Delta \boldsymbol{B}\|_F}{\|\boldsymbol{B}\|_F} = \frac{|k^s - 1|}{\sqrt{k^s + 1}}, \qquad \frac{\langle \boldsymbol{B}, \boldsymbol{B} + \Delta \boldsymbol{B} \rangle}{\|\boldsymbol{B}\|_F \|\boldsymbol{B} + \Delta \boldsymbol{B}\|_F} = \frac{2}{\sqrt{2 + k^s + 1/k^s}}.$$

The proof is completed. $\qquad\qquad\square$

**Proposition 1.** *Let $\boldsymbol{A}$ be a PD matrix of order $m+n$ whose SVD is $\boldsymbol{U\Lambda U}^\mathsf{T}$, where $m, n \in \mathbb{N}_+$, $n = lm$, $\boldsymbol{U} = [\boldsymbol{u}_i]$ is an orthogonal matrix, $\boldsymbol{\Lambda} = \mathrm{diag}([c\lambda \boldsymbol{1}_{m\times 1}^\mathsf{T}, \lambda \boldsymbol{1}_{n\times 1}^\mathsf{T}]^\mathsf{T})$, $c \geq 1000$, and $\lambda > 0$. Given $\Delta \boldsymbol{U} = [\Delta \boldsymbol{u}_i]$, $\Delta \boldsymbol{\Lambda} = \mathrm{diag}([\boldsymbol{0}_{m\times 1}^\mathsf{T}, \Delta \boldsymbol{\lambda}_{n\times 1}^\mathsf{T}]^\mathsf{T})$, and $s \leq -0.25$, we define $\boldsymbol{B} := (\boldsymbol{U\Lambda U}^\mathsf{T})^s$, $\boldsymbol{B}_1 := ((\boldsymbol{U} + \Delta \boldsymbol{U})\boldsymbol{\Lambda}(\boldsymbol{U} + \Delta \boldsymbol{U})^\mathsf{T})^s$, and $\boldsymbol{B}_2 := (\boldsymbol{U}(\boldsymbol{\Lambda} + \Delta \boldsymbol{\Lambda})\boldsymbol{U}^\mathsf{T})^s$. If $\boldsymbol{U} + \Delta \boldsymbol{U}$ is orthogonal, $\|\Delta \boldsymbol{u}_i\|_2 \leq 0.1, \langle \boldsymbol{u}_i, \Delta \boldsymbol{u}_i \rangle \geq -0.005$, $\Delta \boldsymbol{\lambda}_{n\times 1} = (0.02c - 1)\lambda \boldsymbol{1}_{n\times 1}$, and $l = (c/0.02)^s$, then*

$$2\frac{\|\boldsymbol{B}_1 - \boldsymbol{B}\|_F}{\|\boldsymbol{B}\|_F} \leq 0.4 \leq \frac{\|\boldsymbol{B}_2 - \boldsymbol{B}\|_F}{\|\boldsymbol{B}\|_F}, \quad 6\left(1 - \frac{\langle \boldsymbol{B}, \boldsymbol{B}_1 \rangle}{\|\boldsymbol{B}\|_F \|\boldsymbol{B}_1\|_F}\right) \leq 0.06 \leq \left(1 - \frac{\langle \boldsymbol{B}, \boldsymbol{B}_2 \rangle}{\|\boldsymbol{B}\|_F \|\boldsymbol{B}_2\|_F}\right).$$

*Proof.* According to Lemma 1, we have

$$\frac{\|\boldsymbol{B}_1 - \boldsymbol{B}\|_F}{\|\boldsymbol{B}\|_F} \leq 0.2, \qquad \frac{\langle \boldsymbol{B}, \boldsymbol{B}_1 \rangle}{\|\boldsymbol{B}\|_F \|\boldsymbol{B}_1\|_F} \geq (1 - 0.005)^2 \geq 0.99.$$

On the other hand, from Lemma 2(3), we get

$$\frac{\|\boldsymbol{B}_2 - \boldsymbol{B}\|_F}{\|\boldsymbol{B}\|_F} = \frac{|x - 1|}{\sqrt{x + 1}} = f_1(x), \qquad \frac{\langle \boldsymbol{B}, \boldsymbol{B}_2 \rangle}{\|\boldsymbol{B}\|_F \|\boldsymbol{B}_2\|_F} = \frac{2}{\sqrt{2 + x + 1/x}} = f_2(x),$$

where $x = (0.02c)^s \in (0, 20^{-1/4}]$. It is easy to verify that $f_1$ decreases monotonically and $f_2$ increases monotonically for $0 < x < 1$. Hence

$$f_1(x) \geq f_1(20^{-1/4}) \geq 0.4, \quad f_2(x) \leq f_2(20^{-1/4}) \leq 0.94.$$

The proof is completed. $\qquad\qquad\square$

**Lemma 12.** *Let $\{\boldsymbol{X}_t\}_{t=1}^{t=T}$ be a sequence of symmetric matrices, and $\boldsymbol{A}_t = \sum_{s=1}^t \boldsymbol{X}_s$, where $t = 1, \ldots, T$. Suppose we have two sequences of symmetric matrices $\{\boldsymbol{Y}_t\}_{t=1}^{t=T}, \{\boldsymbol{Z}_t\}_{t=0}^{t=T}$, and a sequence real numbers $\{\rho_t\}_{t=0}^{t=T}$ satisfying*

$$\boldsymbol{Y}_t = \boldsymbol{Z}_{t-1} + \boldsymbol{X}_t, \quad \rho_t = \rho_{t-1} + \|\boldsymbol{Y}_t - \boldsymbol{Z}_t\|_2, \quad \boldsymbol{Z}_0 = \boldsymbol{0}, \rho_0 = 0.$$

*Define $\boldsymbol{B}_t = \rho_t \boldsymbol{I} + \boldsymbol{Z}_t$, where $\boldsymbol{I}$ denotes the identity matrix. Then for $t = 1, \ldots, T$, we have*

$$\boldsymbol{B}_t \succeq \boldsymbol{B}_{t-1} + \boldsymbol{X}_t, \quad \boldsymbol{A}_t \preceq \boldsymbol{B}_t \preceq 2\rho_t \boldsymbol{I} + \boldsymbol{A}_t.$$

*Proof.* Note that for any symmetric matrix $\boldsymbol{S}$, it holds that $\|\boldsymbol{S}\|_2\boldsymbol{I} \succeq \boldsymbol{S}$. Then we have

$$(\rho_t - \rho_{t-1})\boldsymbol{I} + \boldsymbol{Z}_t = \|\boldsymbol{Y}_t - \boldsymbol{Z}_t\|_2\boldsymbol{I} + \boldsymbol{Z}_t \succeq \boldsymbol{Y}_t.$$

Adding $\rho_{t-1}\boldsymbol{I}$ on both sides, we get

$$\boldsymbol{B}_t = \rho_t\boldsymbol{I} + \boldsymbol{Z}_t \succeq \rho_{t-1}\boldsymbol{I} + \boldsymbol{Y}_t = \rho_{t-1}\boldsymbol{I} + \boldsymbol{Z}_{t-1} + \boldsymbol{X}_t = \boldsymbol{B}_{t-1} + \boldsymbol{X}_t.$$

Hence

$$\boldsymbol{B}_t = \sum_{s=1}^{t}(\boldsymbol{B}_s - \boldsymbol{B}_{s-1}) \succeq \sum_{s=1}^{t}\boldsymbol{X}_s = \boldsymbol{A}_t.$$

On the other hand, we have

$$\boldsymbol{Z}_t \preceq \|\boldsymbol{Z}_t - \boldsymbol{Y}_t\|_2\boldsymbol{I} + \boldsymbol{Y}_t = (\rho_t - \rho_{t-1})\boldsymbol{I} + \boldsymbol{Y}_t.$$

Adding $\rho_t\boldsymbol{I}$ on both sides, we get

$$\begin{aligned}
\boldsymbol{B}_t = \rho_t\boldsymbol{I} + \boldsymbol{Z}_t &\preceq (2\rho_t - \rho_{t-1})\boldsymbol{I} + \boldsymbol{Y}_t \\
&= 2(\rho_t - \rho_{t-1})\boldsymbol{I} + \rho_{t-1}\boldsymbol{I} + \boldsymbol{Z}_{t-1} + \boldsymbol{X}_t \\
&= \boldsymbol{B}_{t-1} + 2(\rho_t - \rho_{t-1})\boldsymbol{I} + \boldsymbol{X}_t.
\end{aligned}$$

Hence

$$\boldsymbol{B}_t = \sum_{s=1}^{t}(\boldsymbol{B}_s - \boldsymbol{B}_{s-1}) \preceq \sum_{s=1}^{t}2(\rho_s - \rho_{s-1})\boldsymbol{I} + \sum_{s=1}^{t}\boldsymbol{X}_s = 2\rho_t\boldsymbol{I} + \boldsymbol{A}_t.$$

The proof is completed. $\square$

**Theorem 1.** *Assume that the gradients $\boldsymbol{G}_1, \ldots, \boldsymbol{G}_T \in \mathbb{R}^{m\times n}$ are matrices of rank at most $r$. Then for any $\boldsymbol{W}^* \in \mathbb{R}^{m\times n}$ and $\epsilon > 0$, if $\eta = D/\sqrt{2r}$, the regret of Algorithm 6 is bounded as follows,*

$$\sum_{t=1}^{T} f_t(\boldsymbol{W}_t) - \sum_{t=1}^{T} f_t(\boldsymbol{W}^*) \leq \sqrt{2r}D[2^{1/4}m\rho_T^{1/4} + \mathrm{tr}(\tilde{\boldsymbol{L}}_T^{1/4})][2^{1/4}n\mu_T^{1/4} + \mathrm{tr}(\tilde{\boldsymbol{R}}_T^{1/4})],$$

*where $D = \max_{t\in[T]}\|\boldsymbol{W}_t - \boldsymbol{W}^*\|_F$, $\tilde{\boldsymbol{L}}_t = \epsilon\boldsymbol{I}_m + \sum_{t=1}^{T}\boldsymbol{G}_t\boldsymbol{G}_t^{\mathsf{T}}$, and $\tilde{\boldsymbol{R}}_t = \epsilon\boldsymbol{I}_n + \sum_{t=1}^{T}\boldsymbol{G}_t^{\mathsf{T}}\boldsymbol{G}_t$.*

*Proof.* Define $\hat{\boldsymbol{L}}_t = (\epsilon + \rho_t)\boldsymbol{I}_m + \boldsymbol{L}_t$, $\hat{\boldsymbol{R}}_t = (\epsilon + \mu_t)\boldsymbol{I}_n + \boldsymbol{R}_t$. According to Lemma 12, $\hat{\boldsymbol{L}}_t$ and $\hat{\boldsymbol{R}}_t$ are positive definite. Recall the update performed in Algorithm 6,

$$\boldsymbol{W}_{t+1} = \boldsymbol{W}_t - \eta\hat{\boldsymbol{L}}_t^{-1/4}\boldsymbol{G}_t\hat{\boldsymbol{R}}_t^{-1/4}.$$

For $t > 0$, let $\boldsymbol{H}_t = \hat{\boldsymbol{L}}_t^{1/4} \otimes \hat{\boldsymbol{R}}_t^{1/4}$, $\boldsymbol{g}_t = \overline{\mathrm{vec}}(\boldsymbol{G}_t)$ and $\boldsymbol{w}_t = \overline{\mathrm{vec}}(\boldsymbol{W}_t)$. Due to Lemma 3(3) and Lemma 4, we have

$$\boldsymbol{w}_{t+1} = \boldsymbol{w}_t - \eta\boldsymbol{H}_t^{-1}\boldsymbol{g}_t.$$

Lemma 12 implies $0 \prec \hat{\boldsymbol{L}}_1 \preceq \cdots \preceq \hat{\boldsymbol{L}}_T, 0 \prec \hat{\boldsymbol{R}}_1 \preceq \cdots \preceq \hat{\boldsymbol{R}}_T$. Thus, according to Lemma 3(3)(4) and Lemma 6, we get

$$0 \prec \boldsymbol{H}_1 \preceq \cdots \preceq \boldsymbol{H}_T.$$

Let $\boldsymbol{H}_0 = \boldsymbol{0}$. By invoking Lemma 9 and Lemma 8, we obtain the regret bound

$$\begin{aligned}
\sum_{t=1}^{T} f_t(\boldsymbol{W}_t) - \sum_{t=1}^{T} f_t(\boldsymbol{W}^*) &\leq \frac{1}{2\eta}\sum_{t=1}^{T}(\boldsymbol{w}_t - \boldsymbol{w}^*)^{\mathsf{T}}(\boldsymbol{H}_t - \boldsymbol{H}_{t-1})(\boldsymbol{w}_t - \boldsymbol{w}^*) + \frac{\eta}{2}\sum_{t=1}^{T}(\|\boldsymbol{g}_t\|_{\boldsymbol{H}_t}^*)^2 \\
&\leq \frac{D^2}{2\eta}\sum_{t=1}^{T}\mathrm{tr}(\boldsymbol{H}_t - \boldsymbol{H}_{t-1}) + \frac{\eta}{2}\sum_{t=1}^{T}(\|\boldsymbol{g}_t\|_{\boldsymbol{H}_t}^*)^2 \\
&= \frac{D^2}{2\eta}\mathrm{tr}(\boldsymbol{H}_T) + \frac{\eta}{2}\sum_{t=1}^{T}(\|\boldsymbol{g}_t\|_{\boldsymbol{H}_t}^*)^2,
\end{aligned}$$

where $D = \max_{t \in [T]} \|\boldsymbol{w}_t - \boldsymbol{w}^*\|_2 = \max_{t \in [T]} \|\boldsymbol{W}_t - \boldsymbol{W}^*\|_F$ and $\boldsymbol{w}^* = \overline{\mathrm{vec}}(\boldsymbol{W}^*)$.

Define $\widehat{\boldsymbol{H}}_t = (r\epsilon\boldsymbol{I} + \sum_{s=1}^t \boldsymbol{g}_s \boldsymbol{g}_s^\mathsf{T})^{1/2}$. Lemma 11 and Lemma 12 imply that

$$\widehat{\boldsymbol{H}}_t \preceq \sqrt{r}\tilde{\boldsymbol{L}}_t^{1/4} \otimes \tilde{\boldsymbol{R}}_t^{1/4} \preceq \sqrt{r}\boldsymbol{H}_t.$$

Using Lemma 7 and Lemma 10 along with the above equation, we obtain

$$\sum_{t=1}^T (\|\boldsymbol{g}_t\|_{\boldsymbol{H}_t}^*)^2 \leq \sqrt{r}\sum_{t=1}^T (\|\boldsymbol{g}_t\|_{\widehat{\boldsymbol{H}}_t}^*)^2 \leq 2\sqrt{r}\mathrm{tr}(\widehat{\boldsymbol{H}}_T) \leq 2r\mathrm{tr}(\boldsymbol{H}_T).$$

Consequently, using Lemma 3(5) and Lemma 12, we get the desired regret bound

$$\sum_{t=1}^T f_t(\boldsymbol{W}_t) - \sum_{t=1}^T f_t(\boldsymbol{W}^*) \leq \Big(\frac{D^2}{2\eta} + \eta r\Big)\mathrm{tr}(\boldsymbol{H}_T) = \sqrt{2r}D\mathrm{tr}(\hat{\boldsymbol{L}}_T^{1/4})\mathrm{tr}(\hat{\boldsymbol{R}}_T^{1/4})$$

$$\leq \sqrt{2r}D[2^{1/4}m\rho_T^{1/4} + \mathrm{tr}(\tilde{\boldsymbol{L}}_T^{1/4})][2^{1/4}n\mu_T^{1/4} + \mathrm{tr}(\tilde{\boldsymbol{R}}_T^{1/4})],$$

by choosing $\eta = D/\sqrt{2r}$. The proof is completed. $\qquad\square$

## G   Experimental Details

We use one RTX3060Ti GPU under the PyTorch 2.0.1+CUDA11.8 framework for DNN training on the CIFAR-100 and Tiny-ImageNet datasets, use one A800 GPU under the PyTorch 2.0.1+CUDA11.7 framework for DNN training on the ImageNet-1k and C4 datasets, and use two NVIDIA L40S GPUs under the PyTorch 2.0.1+CUDA11.8 framework for DNN training on the OWT dataset. To obtain the total peak memory consumption per GPU, we call "torch.cuda.max_memory_allocated".

We set "torch.backends.cudnn.benchmark" to "False" for all the experiments, except when training ViT-Base/32 on the ImageNet-1k dataset. We report the total memory consumption instead of the memory consumption of the second-order optimizer. This total memory includes data, model parameters, activations, gradients, states forming the preconditioners and their inverse roots, states for the used first-order optimizer, and memory fragments. *Our focus lies in quantizing the states for constructing preconditioners and their inverse roots, which are approximately 7x smaller for 4-bit Shampoo compared to 32-bit Shampoo. Because the block size is 64, its maximum value should be calculated every 64 elements and saved as a 32-bit value, resulting in an additional overhead of 0.5 bits (32/64). Consequently, the memory savings are approximately 7 times, calculated as* $32/(4+0.5)$. In the future, we may adopt double quantization [9] to further reduce memory consumption.

For SGDM, Adagrad or AdamW used in second-order optimizers, we use 32-bit optimizer states on image classification tasks and 16-bit optimizer states on natural language modeling tasks by default. For SGDM, we set the momentum to 0.9 and use an initial learning rate of 0.1. For Adagrad, we set $\epsilon = 10^{-10}$ and use an initial learning rate of 0.01. For AdamW, we set $\beta_1 = 0.9, \beta_2 = 0.999$, and $\epsilon = 10^{-8}$ and use an initial learning rate of 0.001. For quantization settings, we employ block-wise normalization with a block size of 64 and linear square quantization by default. Matrices with a size smaller than 4096 will not be quantized. For Shampoo and CASPR, we use $\epsilon = 10^{-6}, \beta = 0.95$ and $t_1 = 1, t_2 = 4$ by default. Shampoo and CASPR precondition blocks from large matrices and the maximum order of a preconditioner is 10000 for 130M LLAMA-2 and is 1200 for other models. For training loss, we use cross-entropy loss. For image classification tasks, automatic mixed precision is enabled except for training transformers on the CIFAR-100 and Tiny-ImageNet datasets.

**Settings on training CNNs on CIFAR-100 or Tiny-ImageNet.** Minibatch size is set to 128. Weight decay is 0.0005. Data augmentation includes random crop and horizontal flip. For Shampoo, we set $T_1 = 100$ and $T_2 = 500$. In Section 5, we run SGDM for 300 epochs and SGDM+Shampoo for 200 epochs on the CIFAR-100 dataset. We run SGDM for 150 epochs and SGDM+Shampoo for 100 epochs on the Tiny-ImageNet dataset. We adopt the multi-step learning rate schedule (the learning rate is multiplied by 0.1 for every 30% epochs with a linear warmup at the first 5 epochs).

**Settings on training transformers on CIFAR-100 or Tiny-ImageNet.** We set a patch size of 4 for ViT-small on the CIFAR-100 dataset, and a patch size of 8 for ViT-small on the Tiny-ImageNet dataset. For training Swin-Tiny on the CIFAR-100 dataset, we use a patch size of 2 and window size of 4. For training Swin-Tiny on the Tiny-ImageNet dataset, we use a patch size of 4 and window

size of 7. Minibatch size is set to 128. We run Adagrad/AdamW/NadamW for 150 epochs and Adagrad/AdamW+Shampoo for 100 epochs. Weight decay is 0.0005 for Adagrad, and is 0.05 for AdamW/NadamW. We use the cosine learning rate schedule. Data augmentation follows the source code in [25]. For Shampoo, we set $T_1 = 100$ and $T_2 = 500$. With the exception of certain optimizer settings, the configurations used for ablation studies are identical to those outlined above.

**Settings on training ResNet50 on ImageNet-1k.** We run SGDM for 120 epochs and SGDM+Shampoo for 100 epochs. Minibatch size is set to 256. Weight decay is 0.0001. We adopt the multi-step learning rate schedule (the learning rate is multiplied by 0.1 for every 30% epochs with a linear warmup at the first 5 epochs). Data augmentation includes random resized crop, horizontal flip, and color jitter. For Shampoo, we set $T_1 = 200$ and $T_2 = 1000$.

**Settings on training ViT-Base/32 on ImageNet-1k.** We run AdamW for 150 epochs and AdamW+Shampoo for 120 epochs. Minibatch size is set to 512. Weight decay is 0.05. We use the cosine learning rate schedule. Data augmentation follows the configuration for training ViT-Base/16 in [44], excluding repeated augmentation. For Shampoo, we set $T_1 = 200$ and $T_2 = 1000$.

**Settings on training GPT-2 on OWT.** We run AdamW with 10% warmup steps. Total batch size is set to 480. Batch size is set to 24 for training 124M GPT-2. Dtype is bfloat16. Weight decay is 0.1. For Shampoo, we set $T_1 = 200$ and $T_2 = 200$. For our 4-bit Shampoo, we use Schur-Newton iteration used in Algorithm 4 to compute the inverse root of a preconditioner for training stability.

**Settings on training LLAMA-2 on C4.** We run AdamW with 10% warmup steps. Total batch size is set to 512. Batch size is set to 256 for training 130M LLAMA-2 and is set to 128 for training 350M LLAMA-2. Dtype is bfloat16. Weight decay is 0. For Shampoo, we set $T_1 = 200$ and $T_2 = 200$.

**Settings on K-FAC and AdaBK.** K-FAC/AdaBK preconditions layers without limiting the size of a preconditioner. We set $\beta = 0.9$, $T_1 = 200$, and $T_2 = 2000$. We use $\epsilon = 0.1$ for K-FAC and $\epsilon = 0.001$ for AdaBK. For 4-bit K-FAC/AdaBK, we set $t_1 = 0$ and $t_2 = 0$ (i.e., no orthogonal rectification).

**Settings on schedule free optimization.** We use the code from [6] to train ResNet34 with SGDScheduleFree and Swin-Tiny with AdamWScheduleFree. For SGDScheduleFree, we set lr=1.0, weight_decay=0.0005 and warmup_steps=2000. For AdamWScheduleFree, we set lr=0.0025, weight_decay=0.05 and warmup_steps=10000.

**Settings on M-FAC.** We use the code from [15] and set ngrads=32, damp=0.1. The other hyperparameter settings of M-FAC is the same as that of SGDM used for ResNet34 training.

# H    Additional Results

## H.1    Image Classification

**More learning rate schedulers.** Table 8 shows the performance and wall-clock time of training ResNet34 on CIFAR-100 with cosine learning rate decay. By comparison, SGDM+Shampoo still converges faster than SGDM, and have slightly better test performance.

Table 8: Performance and wall-clock time of training ResNet34 on the CIFAR-100 dataset with cosine learning rate decay. TA = test accuracy, and WCT = wall-clock time.

| Epochs | Optimizer | TA (%) | WCT (min) |
|--------|-----------|--------|-----------|
| 200 | SGDM | 79.67 | 116.0 |
| 300 | SGDM | 79.83 | 172.7 |
| 200 | SGDM + 32-bit Shampoo | 80.39 | 152.7 |
| 200 | SGDM + 4-bit Shampoo (our) | 80.22 | 161.7 |

We also provide the results of training ResNet34 and Swin-Tiny on CIFAR-100 with schedule-free approach [6] in Table 9. From it one can see that AdamWScheduleFree achieves comparable performance to AdamW with cosine decay, while SGDScheduleFree underperforms compared to SGDM. We observe that this schedule-free algorithm shows rapid improvements in training and test accuracy during the early training stages, but may fail to achieve a higher test accuracy ultimately (see Figure 9). Anyway, these methods are still worse than our AdamW+4-bit Shampoo.

Table 9: Performance and wall-clock time of training on the CIFAR-100 dataset with cosine learning rate decay and schedule-free approach. ResNet34 is trained for 300 epochs and Swin-Tiny is trained for 150 epochs. TA = test accuracy, and WCT = wall-clock time.

| Model | Optimizer | TA (%) | WCT (min) |
|---|---|---|---|
| ResNet34 | SGDM | 79.83 | 172.7 |
| | SGDScheduleFree | 75.63 | 169.6 |
| Swin-Tiny | AdamW | 76.69 | 318.6 |
| | AdamWScheduleFree | 76.58 | 321.9 |

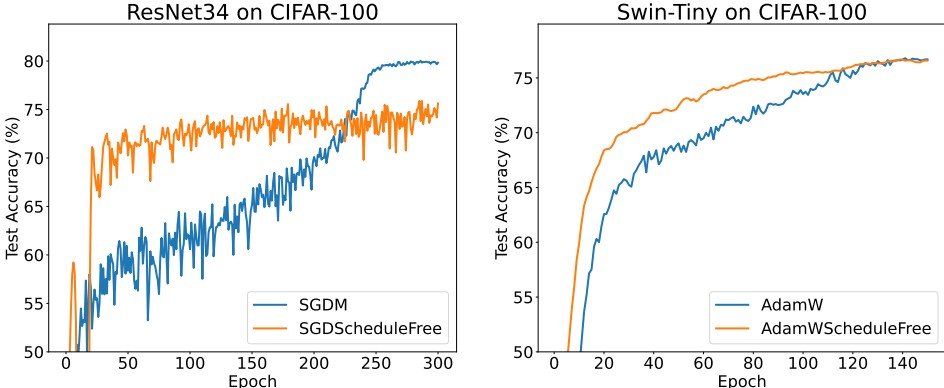

Figure 9: Visualization of test accuracies on the CIFAR-100 dataset with cosine learning rate decay and schedule-free approach.

**More optimizers.** Table 10 shows results of training Swin-Tiny on CIFAR-100 with NadamW, Adagrad and Adagrad+Shampoo. One can see that Adagrad+4-bit Shampoo converges faster than Adagrad with ignorable extra memory overhead, and also has higher test accuracy. Besides, though NadamW [11] is slightly better than AdamW, it is still worse than our AdamW+4-bit Shampoo.

Table 10: Performance, wall-clock time, and memory cost of training Swin-Tiny on the CIFAR-100 dataset. TA = test accuracy, WCT = wall-clock time, and TMC = total GPU memory cost.

| Optimizer | TA (%) | WCT (min) | TMC (MB) |
|---|---|---|---|
| NadamW | 77.11 | 342.4 | 1465.8 |
| AdamW + 32-bit Shampoo | 79.34 | 260.8 | 2036.0 |
| AdamW + 4-bit Shampoo (our) | 78.63 | 273.3 | 1543.9 |
| Adagrad | 66.56 | 294.6 | 1354.9 |
| Adagrad + 32-bit Shampoo | 73.55 | 245.3 | 1930.4 |
| Adagrad + 4-bit Shampoo (our) | 72.66 | 259.6 | 1433.0 |

M-FAC [15] is a matrix-free method computing inverse-Hessian vector products with many gradient copies. It is not memory-efficient for M-FAC to maintain $m$ dense gradient copies ($m = 1024$ in its official code). Table 11 shows that both SGDM+32-bit Shampoo and SGDM+4-bit Shampoo enjoy much higher efficiency than M-FAC ($m = 32$) for training ResNet34 on CIFAR-100, and enjoy higher test accuracy. EVA [42] is a rank-one second-order optimizer and is memory-efficient. We train ResNet34 on CIFAR-100 with SGDM+EVA, but despite extensive hyper-parameter tuning, we fail to achieve acceleration over SGDM. Instead, we cite EVA's result of training VGG-19 on CIFAR-100 for 200 epochs (see Table 2 in [42]). The test accuracies of SGDM+EVA and SGDM+Shampoo are 73% and 74.5%, respectively.

Table 11: Performance and memory cost of training ResNet34 on the CIFAR-100 dataset with cosine learning rate decay. All the optimizers are run for 200 epochs. TA = test accuracy, and TMC = total GPU memory cost.

| Optimizer | SGDM | M-FAC ($m$=32) | SGDM + 32-bit Shampoo | SGDM + 4-bit Shampoo (our) |
|---|---|---|---|---|
| TA (%) | 79.67 | 78.56 | 80.39 | 80.22 |
| TMC (MB) | 822.03 | 3424.8 | 1441.8 | 908.4 |

Table 12: Performance, wall-clock time, and memory usage per GPU on natural language modeling tasks. VL = validation loss, WCT = wall-clock time, and TMC = total GPU memory cost.

| Dataset | Model | Optimizer | VL | WCT (min) | TMC (MB) |
|---|---|---|---|---|---|
| C4 | LLAMA-130M | AdamW | 3.214 | 346.9 | 47026 |
| | | AdamW + 32-bit Shampoo | 3.184 | 353.7 | 48813 |
| | | AdamW + 4-bit Shampoo (naive) | 3.200 | 353.5 | 47316 |
| | | AdamW + 4-bit Shampoo (our) | 3.194 | 353.1 | 47318 |
| | LLAMA-350M | AdamW | 2.939 | 2687 | 54184 |
| | | AdamW + 32-bit Shampoo | 2.908 | 2776 | 59149 |
| | | AdamW + 4-bit Shampoo (naive) | 2.930 | 2753 | 54894 |
| | | AdamW + 4-bit Shampoo (our) | 2.924 | 2795 | 54894 |
| OWT | GPT2-124M | AdamW | 2.954 | 2310 | 27010 |
| | | AdamW + 32-bit Shampoo | 2.936 | 2330 | 28490 |
| | | AdamW + 4-bit Shampoo (naive) | 2.953 | 2359 | 27209 |
| | | AdamW + 4-bit Shampoo (our) | 2.944 | 2311 | 27209 |

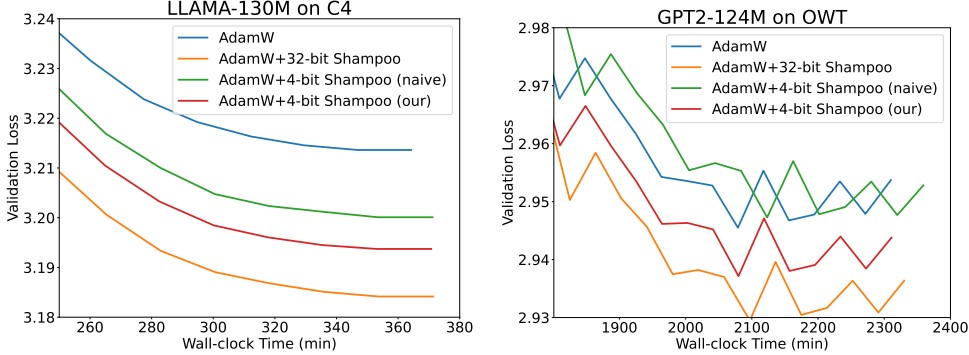

Figure 10: Visualization of validation loss on the C4 and OWT datasets.

## H.2 Natural Language Modeling

**Models, datasets, and hyperparameters.** We train 124M GPT-2 [32] for 60k steps on the Open-WebText (OWT) dataset [*] following the nanoGPT codebase [†] with two NVIDIA L40S GPUs, and train 130M LLAMA-2 [37] for 20k steps and 350M LLAMA-2 for 60k steps on the C4 dataset [33] following [43] with one A800 GPU. See Appendix G for experimental details.

**Main results.** We show the performance, wall-clock time, and memory cost in Table 12, and the validation loss curves in Figure 10. As with the vision tasks, our AdamW+4-bit Shampoo consistently outperformed AdamW and naive AdamW+4-bit Shampoo in terms of performance, and AdamW+32-bit Shampoo in terms of memory usage.

**Memory efficiency.** We further check the memory usage by increasing token batch size for a language model, which is calculated as the batch size multiplied by the context length (see [43]). To train LLAMA2-7B on the C4 dataset using a single A800 GPU (with a maximum memory of 81,920

---

[*] http://Skylion007.github.io/OpenWebTextCorpus.

[†] https://github.com/karpathy/nanoGPT.

MB), we set the context length to 256 and then determine the maximum batch size allowed by each optimizer. For Shampoo, the maximum order of a preconditioner for training LLAMA2-7B is 2048. In all experiments, gradient checkpointing is enabled. Table 13 summarizes the evaluation results. By comparison, the 32-bit Shampoo runs out of memory with a batch size of 2, while our 4-bit Shampoo supports a batch size of 64 for standard training and only encounters memory issues at a batch size of 128. These results clearly demonstrate that our 4-bit Shampoo significantly conserves memory compared to the 32-bit version.

Table 13: Memory cost of training LLAMA2-7B on the C4 dataset with different optimizers. One A800 GPU with a maximum memory of 81,920 MB is enabled. TMC = total GPU memory cost, and OOM = out of memory.

| Optimizer | Batch Size | TMC (MB) |
|---|---|---|
| 8-bit AdamW | 64 | 60135 |
| 8-bit AdamW | 128 | 68689 |
| 8-bit AdamW | 256 | OOM |
| 8-bit AdamW + 32-bit Shampoo | 2 | OOM |
| 8-bit AdamW + 4-bit Shampoo (our) | 64 | 74561 |
| 8-bit AdamW + 4-bit Shampoo (our) | 128 | OOM |

