# OpenReview forum: "4-bit Shampoo for Memory-Efficient Network Training"
_NeurIPS.cc/2024/Conference — NeurIPS 2024 poster_

### Official Review · Reviewer_RMP9 · 2024-06-13

**Soundness:** 3
**Presentation:** 3
**Contribution:** 3
**Rating:** 7
**Confidence:** 4

**Summary:**

Quantization is applied to eigen matrix instead of directly to precond. The eigen matrix is then orthonormalized via  Björck orthonormalization to orthogonalize V.

**Strengths:**

The method seems to regain performance vs just naively quantizeing the preconditioner. This makes Shampoo require about the same about of memory as Adam (at least for the vision experiments in the paper).

They presented the theoretical reason why quantizing the eigen matrix is better and basically showed that the quantization is a form of noise and one can just basically find the closest orthonormalization to recover the proper eigen directions via  Björck orthonormalization.

**Weaknesses:**

Could benefit from extensions to LLMs:

I feel this work lacks experiments/analysis on LLMs. I only say this because the memory needed for 2nd order methods increases a lot more for language models due to their long context window. It is clear that this method would help in that instance but it is not clear to me how much it would help and if this quantization scheme has a performance boost over the naive quantization. In the vision tasks we are training for many epochs, for language we often do not see the same windows over and over again.

Learning rate and Learning rate Schedule:

For ResNet34 CIFAR-100 the model accuracies take a big dip mid training. This is tell tail of a multi-step learning rate schedule (noted on line 514.) Such a schedule is really sub-optimal for SGD+M. This can often result in SGD+M lagging behind other optimizers, which in the paper ultimately lead to Shampoo having a shorter wall-clock time. A linear,cosine, or trapezoidal learning rate decay should be used to give proper performance benchmarks and hence timings. Similar can be said for ResNet-50 on Imagenet-1k.

For the ViT Imagenet 1-k the initial lr should likely be better tuned.

**Questions:**

Would it be possible to re-run resnet experiments with a more proper learning rate decay? This would give more confidence in the timings.
In terms of timing the SoTA right now is nAdamW so Shampoo should be compared to that for wall clock times.

Ideally one would use the very recent schedule free optimization as a benchmark.

https://github.com/facebookresearch/schedule_free

 If it is possible to add these as a benchmarks with timings it would be more convincing.

Furthermore, if one can add a 124M GPT2 model for example from here https://github.com/karpathy/nanoGPT, with timings and memory analysis to consider how the 4bit quantization proposed vs the naive and 32 bit effects performance that would also add a lot to the paper in my opinion.

**Limitations:**

As mentioned above the paper does not consider LLMs which can often have a large impact on memory requirement vs a standard vision models.

The paper does not consider more SoTA optimizers like nAdamW or LARS, and optimization methods like EMA and lookahead.

---

> ### Author Rebuttal · Authors · 2024-08-07
>
> Thank you for the insightful and positive comments. In the following, we provide our point-by-point response and hope our response helps address your concerns. We also look forward to the subsequent discussion which further helps to solve the current issues.
>
> **1) Add experiments on LLMs \& 124M GPT2.**
> Thanks. Due to limited computing resources and the short rebuttal period of 7 days, we trained medium-sized language models, including 124M GPT-2 for 20k steps on the OpenWebText (OWT) dataset using code from nanoGPT (https://github.com/karpathy/nanoGPT), and LLAMA-2 130M for 20k and 350M for 60k steps on the C4 dataset following [1]. For these experiments, we adhered to the exact settings provided in the corresponding papers or GitHub repositories to ensure a fair comparison. The results are reported in Table 2 and Figure 1 of the rebuttal PDF. As with the vision tasks in the manuscript, our AdamW+4-bit Shampoo consistently outperformed AdamW and AdamW+4-bit Shampoo (naive) in terms of performance, and AdamW+32-bit Shampoo in terms of memory usage. Since our algorithm design is not task-specific, we believe its improvements should be consistently achievable on other tasks as well which is also validated by the results on both the computer vision tasks in the manuscript and the natural language processing tasks here.
>
> For LLMs, we have compared our quantization with the naive quantization approach in Table 2 and Figure 1 of the rebuttal PDF. Notably, our 4-bit Shampoo with our quantization outperforms the naive 4-bit Shampoo with naive quantization in terms of validation loss with almost the same training times and memory usage. This demonstrates that the performance gain of our quantization is task-independent.
>
> **2) More proper learning rate decay for ResNet.**
> Thanks. Due to time limitation, we use cosine learning rate decay (initial lr=0.1) to train ResNet34 on CIFAR-100. Experimental results can be found in the table below. By comparison, SGDM+Shampoo still converges faster than SGDM, and have slightly better test performance.
>
> | Epochs |        Optimizer         | TA (%) | WCT (min) |
> | :----: | :----------------------: | :----: | :-------: |
> |  200   |           SGDM           | 79.67  |   116.0   |
> |  300   |           SGDM           | 79.83  |   172.7   |
> |  200   |   SGDM+32-bit Shampoo    | 80.39  |   152.7   |
> |  200   | SGDM+4-bit Shampoo (our) | 80.22  |   161.7   |
>
> **3) Initial learning rate for ViT ImageNet 1-k.**
> Thanks. We use the default initial learning rate provided in [2] to train ViT-Base. Although there may be a better learning rate, the performance reported in our paper is relatively high and reasonable. For ViT-Base/32 on ImageNet 1-k, Table 5 in [3] reports 73.38% accuracy for training 300 epochs, while ours for 150 epochs is 72.87% accuracy.
>
> **4) Comparison with NadamW and schedule free optimization.**
> Per your suggestion, we provide the results of training Swin-Tiny on CIFAR-100 in the table below. We run AdamW, NadamW, and AdamWScheduleFree for 150 epochs and AdamW+Shampoo for 100 epochs. For NadamW, we use its implementation in Timm library, using the same hyperparameters as AdamW. For schedule free optimization, we use the code from [4] to train Swin-Tiny, and set lr=0.0025 (default), weight_decay=0.05, warmup_steps=10000.
>
> |         Optimizer         | TA (%) | WCT (min) |
> | :-----------------------: | :----: | :-------: |
> |           AdamW           | 76.69  |   318.6   |
> |          NadamW           | 77.11  |   342.4   |
> |     AdamWScheduleFree     | 76.58  |   321.9   |
> | AdamW+4-bit Shampoo (our) | 78.63  |   273.3   |
>
> One can see that though improving AdamW, these methods are still worse than our AdamW+4-bit Shampoo.
>
>
>
> [1] Jiawei Zhao, Zhenyu Zhang, Beidi Chen, Zhangyang Wang, Anima Anandkumar, and Yuandong Tian. Galore: Memory-efficient llm training by gradient low-rank projection. In ICML, 2024.
>
> [2] Pan Zhou, Xingyu Xie, and Shuicheng Yan. Win: Weight-decay-integrated Nesterov acceleration for adaptive gradient algorithms. In ICLR, 2023.
>
> [3] Alexey Dosovitskiy, Lucas Beyer, Alexander Kolesnikov, Dirk Weissenborn, Xiaohua Zhai, Thomas Unterthiner, Mostafa Dehghani, Matthias Minderer, Georg Heigold, Sylvain Gelly, Jakob Uszkoreit, and Neil Houlsby. An image is worth 16x16 words: Transformers for image recognition at scale. In ICLR, 2021.
>
> [4] The Road Less Scheduled. arXiv preprint arXiv:2405.15682, 2024.

---

> > ### Comment · Reviewer_RMP9 · 2024-08-08
> >
> > I like it. Thank you for taking the time to do these experiments. As well as to provide clarity on imagenet.
> >
> > LLM looks good, just so I get an intuitive feel for practicality could you guys do a tolerance test for when 4bit quantization stops helping. That is either analyze the LLM theoretically, or just train for a few iterations on the LLM with 4-bit shampoo vs standard shampoo while inverting the precond each iteration to see when Adam cant load into memory, when shampoo cant fit into mem and when 4-bit cant fit into mem. I suspect this will follow closely with the figures in your paper already but would be good to double check as the extended window of LLMs can compilate things.
> >
> > Can you add schedule-free to resnet34 (21M params) as well?
> >
> > On a 8M densenet schedule free is hitting ~78% in 200 epochs so I assume with the extra capacity it might jump up a percent or two.

---

> > > ### Author Response · Authors · 2024-08-11
> > > **Thank you for your prompt and positive feedback.**
> > >
> > > Thank you for your prompt and positive feedback. Below, we provide a point-by-point response to address your concerns.
> > >
> > >
> > >
> > > **1) Check memory usage by increasing token batch size.**
> > >
> > > Thanks. The token batch size for a language model is calculated as the batch size multiplied by the context length (see [1]). To train LLAMA2-7B on the C4 dataset using a single A800 GPU (with a maximum memory of 81,920 MB), we set the context length to 256 and then determined the maximum batch size allowed by each optimizer. For Shampoo, the maximum order of a preconditioner for training LLAMA2-7B is 2048. In all experiments, gradient checkpointing is enabled. The following table summarizes the evaluation results, where "OOM" stands for "out of memory."
> > >
> > > |           Optimizer            | Batch Size | Memory Cost (MB) |
> > > | :----------------------------: | :--------: | :--------------: |
> > > |           8bit AdamW           |     64     |      60135       |
> > > |           8bit AdamW           |    128     |      68689       |
> > > |           8bit AdamW           |    256     |       OOM        |
> > > |   8bit AdamW+32 bit Shampoo    |     2      |       OOM        |
> > > | 8bit AdamW+4 bit Shampoo (our) |     64     |      74561       |
> > > | 8bit AdamW+4 bit Shampoo (our) |    128     |       OOM        |
> > >
> > > By comparison, the 32-bit Shampoo runs out of memory with a batch size of 2, while our 4-bit Shampoo supports a batch size of 64 for standard training and only encounters memory issues at a batch size of 128. These results clearly demonstrate that our 4-bit Shampoo significantly conserves memory compared to the 32-bit version.
> > >
> > >
> > >
> > > **2) Schedule-free method for ResNet34.**
> > >
> > > Thanks. The table below summarizes the results of ResNet34 on CIFAR-100. We run SGDM for both 200 and 300 epochs using cosine learning rate decay. We also run SGDScheduleFree for 200 and 300 epochs using the code from [4], with the default settings: learning rate (lr) of 1.0, weight decay of 0.0005, and 2000 warmup steps. Comparing the results in the following table, it is evident that SGDScheduleFree underperforms compared to SGDM when training ResNet34 on CIFAR-100.
> > >
> > > | Epochs |    Optimizer    | Test Accuracy (%) | Wall-Clock Time (minutes) |
> > > | :----: | :-------------: | :---------------: | :-----------------------: |
> > > |  200   |      SGDM       |       79.67       |           116.0           |
> > > |  300   |      SGDM       |       79.83       |           172.7           |
> > > |  200   | SGDScheduleFree |       74.92       |           117.5           |
> > > |  300   | SGDScheduleFree |       75.63       |           169.6           |
> > >
> > > Indeed, we experimented with various hyperparameter settings for SGDScheduleFree. The following table summarizes the results of training ResNet34 on CIFAR-100 for 100 and 200 epochs (lr=learning rate, wd=weight decay, acc=accuracy). Our observations indicate that SGDScheduleFree shows rapid improvements in training and test accuracy during the early stages of training but ultimately fails to achieve the higher test accuracy.
> > >
> > > |  lr  |   wd    | train acc at 100 epoch | train acc at 200 epoch | test acc at 100 epoch | test acc at 200 epoch |
> > > | :--: | :-----: | :--------------------: | :--------------------: | :-------------------: | :-------------------: |
> > > | 0.1  | 0.0005  |         99.92%         |         99.58%         |        1.000%         |        32.42%         |
> > > | 1.0  | 0.0005  |         95.90%         |         96.87%         |        72.20%         |        74.92%         |
> > > | 1.0  | 0.00005 |         99.58%         |         99.61%         |        1.060%         |        70.63%         |
> > > | 4.0  | 0.0005  |         87.64%         |         88.91%         |        72.34%         |        73.82%         |
> > > | 5.0  | 0.0002  |         93.71%         |         94.47%         |        70.01%         |        74.18%         |
> > > |  10  | 0.0001  |         93.48%         |         94.19%         |        72.90%         |        73.73%         |
> > > | 100  | 0.00002 |         84.49%         |         85.99%         |        74.50%         |        72.15%         |
> > >
> > >
> > >
> > > [1] Jiawei Zhao, Zhenyu Zhang, Beidi Chen, Zhangyang Wang, Anima Anandkumar, and Yuandong Tian. Galore: Memory-efficient llm training by gradient low-rank projection. In ICML, 2024.
> > >
> > > [4] The Road Less Scheduled. arXiv preprint arXiv:2405.15682, 2024.

---

> > > > ### Comment · Reviewer_RMP9 · 2024-08-11
> > > >
> > > > This is exactly what I wanted to see.
> > > >
> > > > I think the paper would benefit from these tables.
> > > >
> > > > "Our observations indicate that SGDScheduleFree shows rapid improvements in training and test accuracy during the early stages of training but ultimately fails to achieve the higher test accuracy."
> > > >
> > > > A accuracy curve for this would also be good for the final paper! I would also note that AdamSchedule free should be used. Kinda weirdly Adam schedule free is actually rsm prop schedule free but that's more a a naming thing.
> > > >
> > > > I would encourage the authors to consider a sharded version of the code if they already have not.
> > > >
> > > > Thank you for your time!
> > > >
> > > > Best wishes

---

> > > > > ### Author Response · Authors · 2024-08-12
> > > > >
> > > > > Many thanks. We are glad our response can address your concerns. We will include the results above and try our best to solve the other issues mentioned. Again, thank you for your insightful and positive comments!!

---

> > > > > > ### Comment · Reviewer_RMP9 · 2024-08-13
> > > > > > **Friendly Notes**
> > > > > >
> > > > > > Great! I noticed some other reviewers talk about EVA, etc if anything you guys should consider CASPR, it gives a better curvature estimate and it should be trivial to extend the work here to CASPR.
> > > > > >
> > > > > > Still, both Shampoo and CASPR require dampening matrices which require more memory.  It would also be interesting to consider applying 4-bit quantization to [PSGD](https://github.com/lixilinx/psgd_torch/tree/master?tab=readme-ov-file#resources) which doesn't require dampening.
> > > > > >
> > > > > > Furthermore, since Kronecker-factorization of these curvatures basically means, per layer curvature, it would be interesting to study the effect of which layer's curvature can be naively quantized without performance drop and which require Björck orthonormalization. This may lead to some reduction in overhead of Björck orthonormalization.
> > > > > >
> > > > > > Best wishes

---

> > > > > > > ### Author Response · Authors · 2024-08-14
> > > > > > >
> > > > > > > Thank you for your valuable comments. Below, we provide a point-by-point response to address your concerns.
> > > > > > >
> > > > > > >
> > > > > > >
> > > > > > > **1) Experiments on CASPR.**
> > > > > > >
> > > > > > > Thanks. We implement CASPR by replacing line 12 of Algorithm 4 and line 14 of Algorithm 3 in our paper: $\hat{G}_t=\hat{L}_tG_t\hat{R}_t$ with
> > > > > > > $$
> > > > > > > J_t=\hat{L}_tG_t + G_t\hat{R}_t; \hat{G}_t=\hat{L}_tJ_t + J_t\hat{R}_t
> > > > > > > $$
> > > > > > > according to Algorithm 1 in [1]. The table below shows the results of training Swin-Tiny on CIFAR-100 for 100 epochs with CASPR vs. Shampoo. From the following results, we can see that Shampoo performs close to CASPR, yet is more time efficient.
> > > > > > >
> > > > > > > |         Optimizer         | Test Accuracy (%) | Wall-Clock Time (minutes) |
> > > > > > > | :-----------------------: | :---------------: | :-----------------------: |
> > > > > > > |    AdamW+32-bit CASPR     |       78.82       |           296.0           |
> > > > > > > |     AdamW+4-bit CASPR     |       78.80       |           317.8           |
> > > > > > > |   AdamW+32-bit Shampoo    |       79.34       |           260.8           |
> > > > > > > | AdamW+4-bit Shampoo (our) |       78.63       |           273.3           |
> > > > > > >
> > > > > > >
> > > > > > >
> > > > > > > **2) About PSGD and dampening matrices.**
> > > > > > >
> > > > > > > Thanks. For our method, it does not need to save a whole dampening matrix in practice, since it can recover dampening matrix by only one scalar. Refer to line 174 of the “low-bit-SO-optimizers/optimizers/shampoo1.py” file in our attached code for details.
> > > > > > >
> > > > > > > Regarding PSGD, we use its official PyTorch implementation (https://github.com/lixilinx/psgd_torch), and report its wall-clock time and memory usage for training ResNet34 on CIFAR-100 for 2 epochs using the psgd.Affine optimizer.  From the results, we can see that psgd.Affine costs more time and memory than SGDM+32 bit Shampoo.
> > > > > > >
> > > > > > > |      Optimizer      | Wall-Clock Time (seconds) | Total GPU memory cost (MB) |
> > > > > > > | :-----------------: | :-----------------------: | :------------------------: |
> > > > > > > |     psgd.Affine     |            539            |           2814.3           |
> > > > > > > |        SGDM         |            103            |           1281.6           |
> > > > > > > | SGDM+32-bit Shampoo |            123            |           1903.3           |
> > > > > > >
> > > > > > >
> > > > > > >
> > > > > > > **3) Per-layer adaptive quantization and orthogonal rectification.**
> > > > > > >
> > > > > > > Thanks. This is an interesting area for exploration, and we plan to address it in future work, as we are unable to complete it within the remaining two days.
> > > > > > >
> > > > > > >
> > > > > > >
> > > > > > > [1] CASPR: Combining Axes Preconditioners through Kronecker Approximation for Deep Learning. In ICLR, 2024.

---

> > > > > > > > ### Comment · Reviewer_RMP9 · 2024-08-14
> > > > > > > >
> > > > > > > > Yes, of course! I did not expect the authors to be able to put together experiments with CASPR or PSGD. Was more of a note for future works.
> > > > > > > >
> > > > > > > > psgd.Affine has the ability to precondition sub-linearlly actually to take less memory than Adam. There is also XMat which is also quite memory efficient. Also timings for psgd are not optimized in torch. There is a jax implementation that seems to be quite a bit faster.
> > > > > > > >
> > > > > > > > psgd + your precond quantization would make a nice combination. psgd is actually quite robust to preconditioner noise. there is theory on this in the 2015 paper.
> > > > > > > >
> > > > > > > > Thank you again for the conversations.

---

> > > > > > > > > ### Author Response · Authors · 2024-08-14
> > > > > > > > >
> > > > > > > > > We appreciate your thoughtful comments and enthusiasm for our work. Your insights will help us further improve our quality. Thank you again for your support.

---

### Official Review · Reviewer_LFix · 2024-07-08

**Soundness:** 4
**Presentation:** 3
**Contribution:** 4
**Rating:** 7
**Confidence:** 5

**Summary:**

The paper presents a way to use a second order optimizer with 4-bit quantization, to reduce memory usage. Second order optimizers such as Shampoo use additional memory to store preconditioners and other variables needed for computing the updates. This extra memory can prevent their usage in training very large models. The authors show that quantifying the Shampoo preconditioners results in poor performance, by comparing the -1/4 power of the preconditioner and its quantized version. They then show that keeping the matrix in its decomposed form A = QSQ' allows one to quantize (and orthonormalize) Q, and this results in preconditioners which are very close to the original preconditioners.

The authors show via experiments that their quantization results in an optimization trajectory that is very close to the unquantized version, while saving up to 40% memory, at the cost of a small runtime overhead.

**Strengths:**

The paper focuses on one aspect of second order optimization, and analyzes it thoroughly. The theoretical justification of the method that the authors provide is intended to provide evidence that this method is better than the naive method. They provide a reasonable set of experiments justifying their work. In addition, the method has the potential of being immediately used in practice.

**Weaknesses:**

The method should help in optimization of large networks, but no results are presented on large networks.
The presentation has some minor issues, noted below.

**Questions:**

1. In table 1, as you noted in the conclusion, please add a row for A (8-bit), to compare with the 4 bit quantization of U.
2. Please add a short paragraph describing DT quantization, to make the presentation self-contained.
3. A graph from x \in [-1,1] to  Q(x) would be nice for each of the quantization schemes.
4. In the Algorithm 3, why do you need separate T_1 and T_2? Since you are computing the SVD at every T_1 steps, it might be not too expensive to also compute the preconditioners at that step.
5. "Singular values are scaled by log 10." --> "Singular values are shown on a log_{10} scale." would be a lot less confusing.

---

> ### Author Rebuttal · Authors · 2024-08-07
>
> Thank you for the insightful and positive comments. In the following, we provide our point-by-point response and hope our response helps address your concerns. We also look forward to the subsequent discussion which further helps to solve the current issues.
>
> **1) Training of large-scale models.**
> Due to the limited computing resources and rebuttal period, we cannot afford sufficient pre-training of very large models. Instead, we train medium-sized language models, including  124M GPT-2 for 20k steps on the OpenWebText (OWT) dataset using the code from https://github.com/karpathy/nanoGPT, and LLAMA-2 130M for 20k and 350M for 60k steps on the C4 dataset following [1].  For these experiments, we adhered to the exact settings provided in the corresponding papers or GitHub repositories to ensure a fair comparison. The results are reported in Table 2 and Figure 1 of the rebuttal PDF. Similar to the vision tasks in the manuscript, our AdamW+4-bit Shampoo consistently outperformed AdamW and AdamW+4-bit Shampoo (naive) in terms of performance, and AdamW+32-bit Shampoo in terms of memory usage.
>
> Additionally, in the manuscript, we have evaluated six models (including VGG, ResNet, ViT, and Swin) on three datasets: CIFAR100, Tiny-ImageNet, and the large-scale ImageNet dataset. These diverse experiments sufficiently demonstrate the superiority of our 4-bit Shampoo over the vanilla 32-bit version.
>
> Finally, our algorithm design does not rely on any specific properties of the tasks, ensuring that the performance improvements are general and transferable. So we believe these improvements should be consistently achievable on other tasks as well. This is validated by the results on both the computer vision tasks in the manuscript and the natural language processing tasks presented in the rebuttal PDF.
>
>
> **2) Add a row for 8-bit A in Table 1.**
> Per your suggestion, we provide the quantization errors of 8-bit quantization schemes in Table 1 of the rebuttal PDF. Comparing Table 1 in the rebuttal PDF with Table 1 in the manuscript, we can see that 4-bit quantization of eigenvector matrix $U$ has smaller quantization errors than 8-bit quantization of the preconditioner $A$. For a clearer comparison, we extract the quantization errors using Linear-2 quantization from the two tables. The results are given in the following table, where $A = A_1$ is derived from the real world as described in Subsection 3.1. We will include these results into the revision.
>
> | Bits | Quantized Matrices | NRE  | AE ($^\circ$) |
> | :--: | :----------------: | :--: | :-----------: |
> | 4 | $A$ | 0.6243 | 17.293 |
> | 4 | $U$ | 0.0543 | 3.1066 |
> | 8 | $A$ | 0.2164 | 7.9751 |
> | 8 | $U$ | 0.0037 | 0.2121 |
>
> **3) Add a description of DT quantization.**
> Thanks. Dynamic tree (DT) quantization for $b$-bit quantization maps $\\{0, 1, \dots, 2^b - 1\\}$ to $\\{0, 1\\}\cup G$, where $G$ is a set of numbers with the following properties: the number in $G$ looks like $\pm q_k \times 10^{-E}$, where
>
> a) $b = 2 + E + F$, where $E, F$ are integers;
>
> b) $q_k = (p_k + p_{k+1}) / 2$, where $k \in \\{0, \dots, 2^F-1\\}$;
>
> c) $ p_j = 0.9j / 2^F + 0.1$, where $j \in \\{0, \dots, 2^F\\}$.
>
> We will add a paragraph in Appendix C for describing quantization maps mentioned in the manuscript.
>
> **4) Add a graph from $x \in [-1,1]$ to $Q(x)$ for each quantization scheme.**
> Per your suggestion, we have drawn the graph from $x \in [-1,1]$ to $Q(x)$ for each quantization scheme, which is similar to Figure 6 in [2]. Unfortunately, the space constraint prevents its inclusion in the one-page rebuttal PDF. We will add this graph in Appendix C.
>
> **5) Why separate $T_1$ and $T_2$ in Algorithm 3.**
> Thanks. We follow previous second-order optimizers [3, 4], and set different values for $T_1$ and $T_2$ ($T_1$=200, $T_2$=2000 in [4]) for more efficient training. This is because the inverse root is much more expensive than preconditioner update, and thus is updated more lazily. Here we keep the default values of $T_1$ and $T_2$ for both 4- and 32-bit Shampoo for a fair comparison.
>
> **6) English improvement.**
> Thank you for your meticulous proofreading. We will make every effort to polish our manuscript and also seek assistance from native English speakers.
>
>
>
> [1] Jiawei Zhao, Zhenyu Zhang, Beidi Chen, Zhangyang Wang, Anima Anandkumar, and Yuandong Tian. Galore: Memory-efficient llm training by gradient low-rank projection. ICML, 2024.
>
> [2] Tim Dettmers, Mike Lewis, Sam Shleifer, Luke Zettlemoyer. 8-bit optimizers via block-wise quantization. In ICLR, 2022.
>
> [3] Rohan Anil, Vineet Gupta, Tomer Koren, Kevin Regan, and Yoram Singer. Scalable second order optimization for deep learning. arXiv preprint arXiv:2002.09018, 2020.
>
> [4] Hongwei Yong, Ying Sun, and Lei Zhang. A general regret bound of preconditioned gradient method for DNN training. In CVPR, 2023.

---

> > ### Comment · Reviewer_LFix · 2024-08-13
> > **Thank you for your reply.**
> >
> > I will retain my score, and look forward to your revised paper.

---

### Official Review · Reviewer_e8nk · 2024-07-12

**Soundness:** 2
**Presentation:** 2
**Contribution:** 2
**Rating:** 5
**Confidence:** 4

**Summary:**

This paper aims to reduce memory usage in second-order optimizers by compressing 32-bit optimizer states to 4-bit. The authors propose a method called 4-bit Shampoo, which quantizes the eigenvector matrix of the preconditioner rather than the preconditioner itself. This approach maintains the performance of 32-bit optimizers while significantly reducing memory requirements.

**Strengths:**

1. This paper introduces an innovative memory reduction technique using 4-bit quantization for second-order optimizers.
2. This paper provides a comprehensive evaluation across various neural network architectures and datasets.
3. This paper includes practical implementation details with a planned release of source code for accessibility.

**Weaknesses:**

1. **Limited Evaluation Scope**:  The experiments are limited to image classification tasks. Evaluating the method on a broader range of tasks, such as natural language processing or reinforcement learning, could provide a more comprehensive assessment of its effectiveness.
2. **Memory Savings Trade-off**: While the method reduces memory usage, the paper does not provide a detailed analysis of the computational overhead introduced by quantization and dequantization processes. Quantization can sometimes introduce additional computation that may offset memory savings.
3. **Quantization Error Handling**: The paper discusses quantization errors and proposes solutions, but it lacks a comprehensive analysis of how these errors impact the overall training dynamics, particularly over long training periods or across different types of neural networks.
4. **Orthogonality Rectification**: The use of Björck orthonormalization to rectify the orthogonality of the quantized eigenvector matrix is an interesting approach, but it may introduce additional computational complexity. The paper should evaluate the impact of this step on the overall efficiency.
5. **Comparison with More Baselines**: The comparison is mainly with 32-bit Shampoo and first-order optimizers. Including more baselines, especially other memory-efficient optimizers like Adagrad[1], M-FAC[2], and EVA[3], would strengthen the validity of the claims.
6. **Lack of Theoretical Framework**:The 4-bit Shampoo paper lacks a solid theoretical foundation, particularly regarding convergence guarantees and the mathematical justification for its quantization approach. Eva, on the other hand, provides a theoretical interpretation from a trust-region optimization perspective.

[1] Duchi J, Hazan E, Singer Y. Adaptive subgradient methods for online learning and stochastic optimization. Journal of machine learning research, 2011, 12(7).

[2] Frantar E, Kurtic E, Alistarh D. M-fac: Efficient matrix-free approximations of second-order information. Advances in Neural Information Processing Systems, 2021, 34: 14873-14886.

[3] Lin Zhang, Shaohuai Shi, Bo Li, Eva: Practical Second-order Optimization with Kronecker-vectorized Approximation, ICML 2023

**Questions:**

Please refer to the section of weaknesses.

**Limitations:**

The authors discussed the limitations of this work on several aspects, such as memory cost of eigenvector matrix, limited scope of tasks, and limited model sizes.

---

> ### Author Rebuttal · Authors · 2024-08-07
>
> Thank you for the insightful and valuable comments.
>
> **1) Evaluation scope.**
> Thanks. We trained medium-sized language models, including 124M GPT-2 and LLAMA-2 130M/350M. The results are reported in Table 2 and Figure 1 of the rebuttal PDF, which are consistent with those of vision tasks. Due to the complexity and limited rebuttal time, we were unable to conduct reinforcement learning experiments. Since our algorithm design is not task-specific, we believe its improvements should be consistently achievable on other tasks as well.
>
> **2) Time cost introduced by quantization and dequantization.**
> Thanks. Block-wise quantization and dequantization are indeed very fast, since they can use the cuda-kernel level parallelism. As a result, they are widely adopted in activation compressed training like [1] and memory efficient optimizers like [2].  As shown in Figure 1(a), the quantization and dequantization in AdamW+4-bit Shampoo (naive) only bring around 3% extra overhead, while greatly improving memory efficiency by 24%.
>
> **3) Quantization error analysis over training.**
> Thanks. Quantization-based approaches always try to reduce the quantization errors compared to their full-precision counterparts, so as to expect similar performance. Quantization errors will accumulate along with the training, but they are controllable. As evident from the training curves in Figures 1 and 4, no jump occurs. We experimentally analyzed the impact of quantization errors on the training process (see Appendix D2). The quantization errors of our 4-bit Shampoo are lower than those of naive Shampoo in most of cases and these errors are all bounded.
>
> **4) Time cost introduced by orthogonality rectification.**
> Thanks. We follow the vanilla Shampoo method and perform orthogonality rectification (OR) every $T_1$ or $T_2$ iterations ($T_1$ = 100/200, $T_2$ = 500/1000). See line 5 and line 9 in Algorithm 3. This lazy update strategy greatly reduces the computational cost. We trained ResNet34 on CIFAR-100 with 4-bit Shampoo, both with and without OR, resulting in training times of 161.7 and 161.0 minutes, respectively, with OR introducing less than 1% extra overhead.
>
> **5) Comparison with Adagrad, M-FAC, and EVA.**
> Thanks.  For Adagrad, it is not as widely used as AdamW due to its inferior performance (e.g. see Table 7 in [2]). We integrate Shampoo with Adagrad, and run Adagrad for 150 epochs and Adagrad+Shampoo for 100 epochs. The table below shows that when training Swin-Tiny on CIFAR-100,  Adagrad + 4 bit Shampoo converges faster than Adagrad with ignorable memory overhead, and also has higher test accuracy.
>
> |           Optimizer           | TA (%) | WCT (min) | TMC (MB) |
> | :---------------------------: | :----: | :-------: | :------: |
> |            Adagrad            | 66.56  |   294.6   |  1354.9  |
> |   Adagrad + 32 bit Shampoo    | 73.55  |   245.3   |  1930.4  |
> | Adagrad + 4 bit Shampoo (our) | 72.66  |   259.6   |  1433.0  |
>
> For M-FAC, the following table shows that our SGDM+4bit Shampoo enjoys much higher efficiency than M-FAC (m=32) on ResNet34, and enjoys higher test accuracy. This is because M-FAC needs to maintain m dense gradient copies (m=1024 in its official code), and is not memory-efficient.  Here we run all the optimizers for 200 epochs. One can observe that even M-FAC using m=32 already requires much more GPU memory, let alone M-FAC using m=1024, and also suffers from worse performance than SGDM and SGDM+32/4 bit Shampoo.
>
> | Optimizer |   SGDM    |   M-FAC   | SGDM +32 bit Shampoo | SGDM +4bit Shampoo (our) |
> | :-------: | :-------: | :-------: | :------------------: | :----------------------: |
> |  Memory   | 822.03 MB | 3424.8 MB |      1441.8 MB       |        908.40 MB         |
> | Accuracy  |  79.67%   |  78.56%   |        80.39%        |          80.22%          |
>
> Regarding EVA, it is a rank-one second-order optimizer and is memory-efficient. We trained ResNet34 on CIFAR-100 with SGDM+EVA, but despite extensive hyper-parameter tuning, we failed to achieve acceleration over SGDM. Instead, we cited EVA's result of training VGG-19 on CIFAR-100 for 200 epochs (see Table 2 in [5]). The test accuracies of SGDM+EVA and SGDM+Shampoo are 73% and 74.5%, respectively.
>
> **6) Theoretical analysis, e.g., convergence guarantees \& mathematical justification for quantization.**
>
> Regarding convergence, we do not provide an explicit proof in this work, as our primary focus is on the practical application of optimizers to reduce GPU memory usage. This approach aligns with many other works on optimizers that prioritize practical solutions over theoretical analysis, such as [2][4].
>
> However, we find that the proof techniques of Theorem 1 in [3] can be adapted to demonstrate the convergence of our optimizer. Specifically, we can generalize Lemma 2 used in proving Theorem 1 in [3] and then use this extended lemma to prove the convergence of quantized Shampoo. For further details, please refer to  Lemma 1 in the "global” response. Note that EVA does not prove convergence, but rather demonstrates that its update step is more aggressive than that of K-FAC from the trust region optimization perspective.
>
>
>
> In terms of quantization, we do provide a mathematical justification in Section 4 of our manuscript. We observe that the eigenvalues of Shampoo's preconditioner decrease very rapidly. So we assume that the eigenvalues follow a certain distribution, and theoretically analyze why quantizing the eigenvector matrix of the preconditioner outperforms direct preconditioner quantization.
>
> [1] Division: memory efficient training via dual activation precision. ICML, 2023.
>
> [2] 8-bit optimizers via block-wise quantization. ICLR, 2022.
>
> [3] Block low-rank preconditioner with shared basis for stochastic optimization. NeurIPS, 2023.
>
> [4] Adafactor: Adaptive learning rates with sublinear memory cost. ICML, 2018.
>
> [5] Eva: Practical second-order optimization with Kronecker-vectorized approximation. ICLR, 2023.

---

> > ### Comment · Reviewer_e8nk · 2024-08-12
> > **Response to the rebuttal**
> >
> > Thanks for the reply. My main concern is the insufficient comparison with Adagrad, M-FAC, and EVA. The authors have addressed it to some extent, so I would like to raise my score.

---

> > > ### Author Response · Authors · 2024-08-12
> > >
> > > We appreciate the reviewer's prompt feedback and the increased score. We will include the comparisons with Adagrad, M-FAC, and EVA in the revision.

---

### Official Review · Reviewer_b2wM · 2024-07-14

**Soundness:** 4
**Presentation:** 4
**Contribution:** 3
**Rating:** 6
**Confidence:** 3

**Summary:**

This work introduces a quantized Shampoo method aimed at memory-efficient network training. Shampoo, as a second-order optimizer, incurs additional memory demands due to its optimization states. By implementing 4-bit quantization specifically on Shampoo, this work effectively reduces memory usage. The primary innovation involves quantizing the eigenvector matrix of a preconditioner instead of the preconditioner itself, which significantly minimizes quantization errors. Additionally, this work incorporates Bjorck orthonormalization to further enhance performance. Extensive experiments across multiple models and datasets validate the effectiveness of the proposed methods.

**Strengths:**

- This study provides both empirical and theoretical justification for the quantization approach, demonstrating that quantizing the eigenvector matrix of a preconditioner significantly reduces quantization error.
- The experiments span various datasets and models, consistently achieving performance that matches that of the full-precision Shampoo.
- Both the speed and memory costs are evaluated in the experiments.

**Weaknesses:**

I find this work to be both sound and interesting. One suggestion for improvement is to expand the reporting of memory costs beyond the total GPU memory consumption. Providing a breakdown of memory usage would clarify how the 4-bit Shampoo achieves memory-efficient training.

**Questions:**

Please refer to the weakness.

**Limitations:**

The authors have effectively addressed several limitations, including the efficiency sacrifices due to the non-symmetric properties of eigenvector results, the evaluation being limited to image classification tasks, and the focus on only small-scale models.

---

> ### Author Rebuttal · Authors · 2024-08-07
>
> Thank you for the insightful and positive comments. In the following, we provide our point-by-point response and hope our response helps address your concerns. We also look forward to the subsequent discussion which further helps to solve the current issues.
>
> **1)  Breakdown of memory usage.**
> Thanks. Per your suggestion, in the table below, we have reported the additional memory cost incurred by injecting the vanilla 32-bit Shampoo and our proposed 4-bit Shampoo into the vanilla optimizer. The extra memory cost primarily comes from the preconditioning matrices L and R, and their inverse roots. See Eqn. (1).
>
> To compute this extra memory cost, we use the data from Table 2 of manuscript, and subtract the total memory cost (TMC) of the vanilla optimizer "A" from the TMC of "A + 32/4-bit Shampoo", where A can be SGDM and AdamW. Comparatively, the extra memory cost of the 32-bit Shampoo is seven times greater than that of our 4-bit Shampoo, demonstrating the superior efficiency of our 4-bit Shampoo. This high efficiency is also analyzed and emphasized in Appendix F.
>
>
> |   Dataset   |    Model    |          Optimizer          | Memory Cost |
> | :---------: | :---------: | :-------------------------: | :---------: |
> |  CIFAR-100  |  ResNet34   |    SGDM + 32-bit Shampoo    |  619.8 MB   |
> |  CIFAR-100  |  ResNet34   | SGDM + 4-bit Shampoo (our)  |  86.37 MB   |
> |  CIFAR-100  |  ViT-Small  |   AdamW + 32-bit Shampoo    |  532.0 MB   |
> |  CIFAR-100  |  ViT-Small  | AdamW + 4-bit Shampoo (our) |  71.70 MB   |
> | ImageNet-1k |  ResNet50   |    SGDM + 32-bit Shampoo    |  630.2 MB   |
> | ImageNet-1k |  ResNet50   | SGDM + 4-bit Shampoo (our)  |  89.03 MB   |
> | ImageNet-1k | ViT-Base/32 |   AdamW + 32-bit Shampoo    |   1534 MB   |
> | ImageNet-1k | ViT-Base/32 | AdamW + 4-bit Shampoo (our) |  204.1 MB   |

---

### Author Rebuttal · Authors · 2024-08-07

We thank all the reviewers for their valuable comments and suggestions. In the attached PDF, we have included additional results to address the reviewers' concerns, specifically in Table 1, Table 2, and Figure 1. We also give Lemma 1 to prove the convergence of our 4-bit Shampoo. We will incorporate these results and lemma in the appendix of the revised manuscript.

**Table 1:**
As mentioned in the conclusion section of the manuscript, preconditioners in Shampoo are symmetric matrices and thus can be stored as upper triangular matrices, saving almost half of the memory usage. However, the eigenvector matrix of a preconditioner is not symmetric, resulting in an 8-bit preconditioner occupying the same memory as its 4-bit eigenvector matrix. Comparing Table 1 in the attached PDF with Table 1 in the manuscript, one can observe that the 4-bit quantization of the eigenvector matrix $U$ has smaller quantization errors than the 8-bit quantization of the preconditioner $A$. For a clearer comparison, we extract the quantization errors using Linear-2 quantization from the two tables. The results are given in the following table, where $A = A_1$ is derived from real-world data as described in Subsection 3.1 of the manuscript.

| Bits | Quantized Matrices | NRE  | AE ($^\circ$) |
| :--: | :----------------: | :--: | :-----------: |
| 4 | $A$ | 0.6243 | 17.293 |
| 4 | $U$ | 0.0543 | 3.1066 |
| 8 | $A$ | 0.2164 | 7.9751 |
| 8 | $U$ | 0.0037 | 0.2121 |

**Table 2 and Figure 1:**
In the manuscript, we provided diverse experiments on computer vision tasks to demonstrate that the performance improvements of our approach are general and transferable. Due to limited computing resources and the short rebuttal period of 7 days, we trained medium-sized language models to address the reviewers' concerns. Specifically, we trained a 124M GPT-2 for 20k steps on the OpenWebText (OWT) dataset using code from nanoGPT (https://github.com/karpathy/nanoGPT), and LLAMA-2 130M for 20k and 60k steps on the C4 dataset following [1]. Unless otherwise noted, the maximum order of a preconditioner for training GPT-2 is 1200, and for training LLAMA-2 is 10000. As with the vision tasks in the manuscript, our AdamW+4-bit Shampoo consistently outperformed AdamW and AdamW+4-bit Shampoo (naive) in terms of performance, and AdamW+32-bit Shampoo in terms of memory usage. Since our algorithm design is not task-specific, we believe its improvements should be consistently achievable on other tasks as well.

**Lemma 1:**
We find that the proof techniques of Theorem 1 in [2] can be adapted to demonstrate the convergence of our optimizer. Specifically, we can generalize Lemma 2 used in proving Theorem 1 in [2] and then use this extended lemma to prove the convergence of quantized Shampoo. Our extension of Lemma 2 in [2] is the Lemma 1  below. Providing a complete and detailed convergence analysis within a short 7-day rebuttal period is challenging, but we will make every effort to include this analysis in the revised manuscript.

**Lemma 1.**  Let $\\{X_t\\}$ be a sequence of symmetric matrices, and $A_t=\sum_{s=1}^tX_s$, where $t=1, \dots, T$. Suppose we have two sequences of symmetric matrices $\\{Y_t\\}$, $\\{ Z_t \\}$, and a sequence of real numbers $\\{\rho_t\\}$ satisfying
$$
Y_t = Z_{t-1}+X_t, \quad \rho_t=\rho_{t-1}+\|\| Y_t - Z_t\|\|, \quad Z_0=0, \rho_0=0.
$$
Define $B_t=\rho_tI+Z_t$, where $I$ denotes the identity matrix. Then for $t=1, \dots, T$, we have
$$
B_t \succeq B_{t-1} + X_t, \quad A_t \preceq B_t \preceq 2\rho_tI + A_t.
$$
*Proof.*  Note that for any symmetric matrix $S$, it holds that $\|\|S\|\|I\succeq S$. Then we have
$$
(\rho_t-\rho_{t-1})I + Z_t = \|\| Y_t - Z_t\|\|I + Z_t \succeq Y_t.
$$
Adding $\rho_{t-1}I$ on both sides, we get
$$
B_t = \rho_tI + Z_t \succeq \rho_{t-1}I + Y_t = \rho_{t-1}I + Z_{t-1}+X_t = B_{t-1} + X_t.
$$
Hence
$$
B_t = \sum_{s=1}^{t}(B_s-B_{s-1}) \succeq \sum_{s=1}^{t}X_s = A_t.
$$
On the other hand, we have
$$
Z_t \preceq \|\| Z_t - Y_t\|\|I + Y_t = (\rho_t-\rho_{t-1})I + Y_t.
$$
Adding $\rho_t I$ on both sides, we get
$$
B_t  = \rho_t I + Z_t \preceq (2\rho_t-\rho_{t-1})I + Y_t
$$
$$
		 = 2(\rho_t-\rho_{t-1})I + \rho_{t-1}I + Z_{t-1}+X_t
$$
$$
		 = B_{t-1} + 2(\rho_t-\rho_{t-1})I + X_t.
$$
Hence
$$
B_t = \sum_{s=1}^{t}(B_s-B_{s-1}) \preceq \sum_{s=1}^{t}2(\rho_s-\rho_{s-1})I + \sum_{s=1}^{t}X_s = 2\rho_tI+ A_t.
$$
The proof is completed.

[1] Jiawei Zhao, Zhenyu Zhang, Beidi Chen, Zhangyang Wang, Anima Anandkumar, and Yuandong Tian. Galore: Memory-efficient llm training by gradient low-rank projection. In ICML, 2024.

[2] Jui-Nan Yen, Sai Surya Duvvuri, Inderjit S. Dhillon, and Cho-Jui Hsieh. Block low-rank preconditioner with shared basis for stochastic optimization. NeurIPS, 2023.

---

### Decision · Program_Chairs · 2024-09-25

**Decision:**

Accept (poster)

**Comment:**

Shampoo+Adam demonstrates improved performance compared to vanilla Adam. However, its substantial memory consumption poses a challenge for large-scale training applications. This paper is pioneering in introducing 4-bit quantization to Shampoo, significantly reducing its memory requirements. Specifically, it innovatively quantizes the eigenvector matrix of the preconditioner rather than the preconditioner itself, thereby minimizing quantization errors. Additionally, the paper cleverly incorporates Bjorck orthonormalization to correct orthogonality, further enhancing performance.

All reviewers unanimously agreed on accepting this work. After personally reading the paper and the comments, I also recommend its acceptance due to its novelty and practicality, but I still have a concern that may need to be addressed in future work. Shampoo+Adam requires approximately 50% more memory than vanilla Adam, and this may limit its practical application in training large models. Moreover, while vanilla Adam’s memory usage can be significantly reduced using the  technique of ZeRO during LLM training, it appears challenging to apply ZeRO to Shampoo due to the need to involve all elements of gradients to compute the preconditioners.